# Human embryoid bodies as a novel system for genomic studies of functionally diverse cell types

Katherine Rhodes[1†], Kenneth A Barr[1†], Joshua M Popp[2], Benjamin J Strober[2], Alexis Battle[2,3,4*], Yoav Gilad[1*]

[1]Department of Medicine, University of Chicago, Chicago, United States; [2]Department of Biomedical Engineering, Johns Hopkins University, Baltimore, United States; [3]Department of Computer Science, Johns Hopkins University, Baltimore, United States; [4]Department of Genetic Medicine, Johns Hopkins University, Baltimore, Maryland, United States

**Abstract** Practically all studies of gene expression in humans to date have been performed in a relatively small number of adult tissues. Gene regulation is highly dynamic and context-dependent. In order to better understand the connection between gene regulation and complex phenotypes, including disease, we need to be able to study gene expression in more cell types, tissues, and states that are relevant to human phenotypes. In particular, we need to characterize gene expression in early development cell types, as mutations that affect developmental processes may be of particular relevance to complex traits. To address this challenge, we propose to use embryoid bodies (EBs), which are organoids that contain a multitude of cell types in dynamic states. EBs provide a system in which one can study dynamic regulatory processes at an unprecedentedly high resolution. To explore the utility of EBs, we systematically explored cellular and gene expression heterogeneity in EBs from multiple individuals. We characterized the various cell types that arise from EBs, the extent to which they recapitulate gene expression in vivo, and the relative contribution of technical and biological factors to variability in gene expression, cell composition, and differentiation efficiency. Our results highlight the utility of EBs as a new model system for mapping dynamic inter-individual regulatory differences in a large variety of cell types.

**\*For correspondence:**
ajbattle@jhu.edu (AB);
gilad@uchicago.edu (YG)

†These authors contributed equally to this work

## Editor's evaluation

The authors generated embryoid bodies (EBs) from induced pluripotent stem cells (iPSCs) using a strong mixed-pool study design and performed scRNA-seq profiling. From this data, they identify dozens of cell types and infer differentiation trajectories that align well with known developmental gene expression dynamics. This system is likely to be a good platform for larger eQTL studies that interrogate new cell states.

## Introduction

Genome-wide association studies (GWAS) have identified thousands of genetic variants associated with human traits and diseases, many of which are located in noncoding regions of the genome and are putatively regulatory in function (*Albert and Kruglyak, 2015*). To understand regulatory and functional effects of trait-associated variants, it is necessary to perform molecular assays in the relevant cell types at the relevant stages of development, and potentially to also model different environmental exposures (*Umans et al., 2020*).

**eLife digest** One major goal of human genetics is to understand how changes in the way genes are regulated affect human traits, including disease susceptibility. To date, most studies of gene regulation have been performed in adult tissues, such as liver or kidney tissue, that were collected at a single time point. Yet, gene regulation is highly dynamic and context-dependent, meaning that it is important to gather data from a greater variety of cell types at different stages of their development. Additionally, observing which genes switch on and off in response to external treatments can shed light on how genetic variation can drive errors in gene regulation and cause diseases.

Stem cells can produce more cells like themselves or differentiate – acquire the characteristics – of many cell types. These cells have been used in the laboratory to research gene regulation. Unfortunately, these studies often fail to capture the complex spatial and temporal dynamics of stem cell differentiation; in particular, these studies are unable to observe gene regulation in the transient cell types that appear early in embryonic development. To overcome these limitations, scientists developed systems such as embryoid bodies: three-dimensional aggregates of stem cells that, when grown under certain conditions, spontaneously develop into a variety of cell types.

Rhodes, Barr et al. wanted to assess the utility of embryoid bodies as a model to study how genes are dynamically regulated in different cell types, by different individuals who have distinct genetic makeups. To do this, they grew embryoid bodies made from human stem cells from different individuals to examine which genes switched on and off as the stem cells that formed the embryoid bodies differentiated into different types of cells. The results showed that it was possible to grow embryoid bodies derived from genetically distinct individuals that consistently produce diverse cell types, similar to those found during human fetal development.

Rhodes, Barr et al.'s findings suggest that embryoid bodies are a useful model to study gene regulation across individuals with different genetic backgrounds. This could accelerate research into how genetics are associated with disease by capturing gene regulatory dynamics at an unprecedentedly high spatial and temporal resolution. Additionally, embryoid bodies could be used to explore how exposure to different environmental factors during early development affect disease-related outcomes in adulthood in different individuals.

However, most efforts to identify genetic variants that regulate gene expression (expression quantitative trait loci, or eQTLs) have relied on adult tissue samples collected at a single time point. While such efforts have mapped millions of static, steady-state eQTLs across dozens of tissues and cell types (*GTEx Consortium, 2020*), most disease-associated variants were not found to also be classified as eQTLs (*Yao et al., 2020*; *Aguet et al., 2017*).

It is possible that dynamic and variable regulatory genetic effects, including those that are specific to a given cell type, time point, or environment, may underlie the mechanisms for many unexplained phenotypic associations. For example, recent efforts to characterize gene regulatory dynamics in human induced pluripotent stem cells (iPSCs) and their derived cell types have identified dynamic eQTLs that are associated with disease risk, supporting the intuitive notion that changes in gene regulation during development may play a role in shaping human adult phenotypes, including disease (*Strober et al., 2019*; *Cuomo et al., 2020*). Still, iPSCs are limited in their potential for identifying dynamic regulatory effects. The number of cell types that can be obtained from iPSCs using directed differentiation protocols is quite modest, and time-course experiments, although useful for studying gene regulation at discrete points along a continuum, are inefficient, expensive, and laborious to perform.

With these challenges in mind, we wanted to develop and characterize a new in vitro model for studying gene regulation – a model capable of measuring gene expression continuously along the developmental trajectories towards multiple cell types, and to be able to do so in multiple individuals. To do this, we used iPSCs to form embryoid bodies (EBs), which are three-dimensional aggregates of spontaneously and asynchronously differentiating cells. EB formation has been used to verify stem cell pluripotency for decades; yet, until recently, the complexity of EB cellular composition has precluded their use in genomic studies. With single-cell RNA-sequencing (scRNA-seq), it is now possible to characterize the numerous spatially and developmentally distinct cell types within EBs, including transient

cell types that would otherwise be inaccessible. Indeed, recent scRNA-seq studies of human EB differentiation have revealed the diversity of cell types composing these structures and the transcriptional dynamics governing early fate decisions (*Han et al., 2018*; *Guo et al., 2019*).

To date, however, the only studies that have sequenced EB cells have relied on a small sample of cells from a single individual, leaving a gap in our understanding of technical, biological, and inter-individual variation present in this system (*Han et al., 2018*; *Guo et al., 2019*). Understanding the sources of variation that affect cell composition and scRNA-seq data from EBs is crucial for evaluating the utility of EBs as a novel system for population-level studies of gene regulation. To this end, we used a batch-controlled study design to generate and sequence EBs from multiple individuals using multiple replicates. This allowed us to measure the degree of technical and biological variability in cell identity and gene expression levels associated with repeated independent EB differentiations. We evaluated the consistency in cell type composition across replicates and individuals, characterized the structure of variation in gene expression across the entire data set, and finally, captured patterns of dynamic gene expression along distinct developmental trajectories. Our results indicate that scRNA-seq of differentiating EBs has the potential to be a powerful model system for the study of inter-individual variation in gene regulation across an array of functionally and temporally diverse cell types.

## Results

We have performed a pilot study to establish and characterize the EB system. Toward the ultimate goal of performing dynamic eQTL studies using EBs, we designed a study that allowed us to effectively estimate different sources of variation in single-cell data from EBs. In this pilot study, we focused on consistency of the non-directed differentiation process, and the proportion of gene expression variability that can be explained by technical or biological factors.

### Study Design, data collection, and preprocessing

To characterize sources of variation in gene expression in human EBs, we initially differentiated EBs from three human iPSC lines (18511, 18858, and 19160) in three replicates (see Materials and methods). We performed the experiment in three batches, where each batch includes one replicate from each of the three individuals. EBs differentiate quickly, with cell types representing endoderm, mesoderm, and ectoderm present after 8 days (*Han et al., 2018*). In this study, we maintained EBs for 3 weeks after formation, allowing cells to continue to differentiate and mature. After 21 days, we collected scRNA-seq data, targeting equal numbers of cells from each individual in each replicate. After filtering and quality control (Materials and methods), we retained high-quality data from a sample of 42,488 cells (an average of 4721 cells per individual/replicate). For these cells, we obtained a median of 16,712 UMI counts per cell, which allowed us to measure the expression of a median of 4274 genes per cell (*Figure 1—figure supplement 1*). We integrated data from all cells using Harmony, which anchors the data sets by cell type (*Korsunsky et al., 2019*).

After these initial collections, we found that one individual, 18858, had lower differentiation efficiency than the other two lines (*Figure 1G*, see Differentiation efficiency across individuals). Too assess the robustness of differentiation efficiency and cell type composition among a larger sample of individuals from our YRI iPSC panel, we differentiated EBs in one replicate from each of five additional randomly chosen lines (18856, 18912, 19140, 19159, and 19210). After filtering and quality control, we retained an average of 5243 cells per individual in this new set of data, a median of 5983 UMI counts per cell, and a median of 2775 genes per cell (*Figure 1—figure supplement 4*). Throughout the text, when we use data from the five lines that were subsequently collected using only a single replicate (18856, 18912, 19140, 19150, and 19210), we refer to them as the 'additional lines'. The initial replicated data collected from individuals 18511, 18858, and 19160 are used in all analyses throughout this study, while the additional lines are used only to demonstrate consistency of cell type composition across individuals.

### Cell type composition

To validate our expectation that EBs should contain cells from each germ layer, we first characterized the expression of early developmental marker genes. We found cells expressing markers for

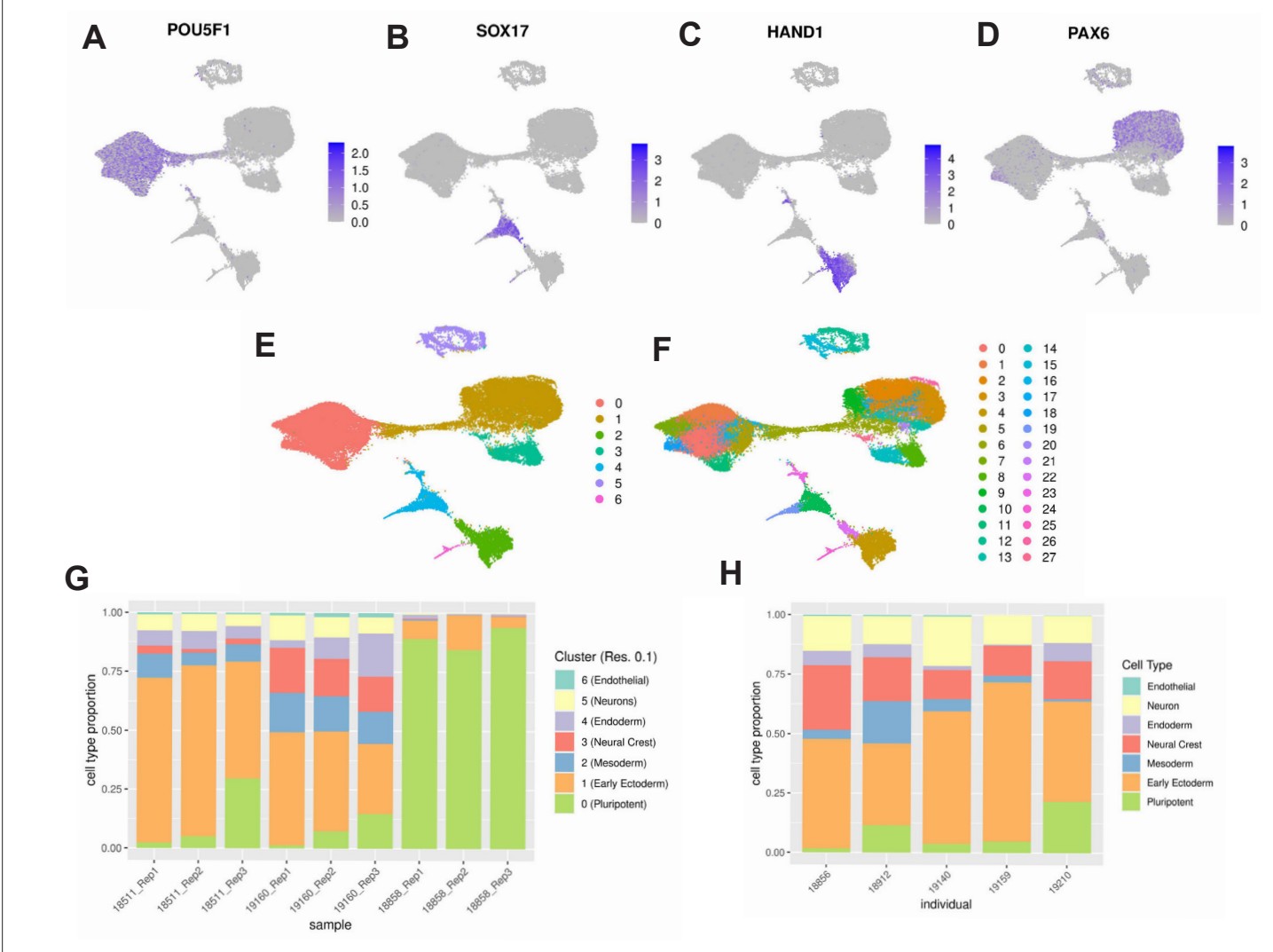

**Figure 1.** Characterization of EB cell type composition using marker gene expression and clustering. (**A–F**) Visualization of EB cells with UMAP. (**A**) Cells from lines 18511, 18858, and 19160 colored by expression of pluripotent marker gene POU5F1, (**B**) Cells from lines 18511, 18858, and 19160 colored by expression of endoderm marker gene SOX17, (**C**) Cells from lines 18511, 18858, and 19160 colored by expression of mesoderm marker gene HAND1, (**D**) Cells from lines 18511, 18858, and 19160 colored by expression of early ectoderm marker gene PAX6. In A-D cells are colored by normalized counts. (**E**) Cells from lines 18511, 18858, and 19160 colored by Seurat cluster assignment at clustering resolution 0.1. (**F**) Cells from lines 18511, 18858, and 19160 colored by Seurat cluster assignment at clustering resolution 1. (**G**) Proportions of cells from replicates of lines 18511, 18858, and 19160 assigned to Seurat clusters at clustering resolution 0.1. (**H**) Proportions of cells from additional lines assigned to broad cell types present in EBs.

The online version of this article includes the following figure supplement(s) for figure 1:

**Figure supplement 1.** Quality metrics after filtering.

**Figure supplement 2.** Seurat clusters identified at clustering resolution 0.5 (Left) and 0.8 (Right).

**Figure supplement 3.** UMAP visualization of cells from individual 18858 only.

**Figure supplement 4.** Cell type composition of additional YRI lines.

endoderm (*SOX17*, *FOXA2*), mesoderm (*HAND1*), and ectoderm (*PAX6*), in addition to cells still expressing pluripotency markers (*POU5F1*, *MYC*, *NANOG*). We visualized the data with uniform manifold approximation and projection (UMAP) and observed that cells expressing each of these germ layer markers occupied distinct groups in UMAP space (*Figure 1A–D*; *Becht et al., 2018*). Moreover, we found that every replicate in our experiment, regardless of the individual, includes cells from all three germ layers (*Figure 1E and G*).

We next sought to further explore the heterogeneous cell types present in these EBs. In studies of scRNA-seq data from tissues and samples with well-characterized cell type composition, clustering is often applied to demarcate populations of pure cell types within heterogeneous samples. In these studies, clustering resolution, which determines the number of clusters identified by the algorithm, is typically chosen to recapitulate the expected number of known cell types. The identified clusters can be annotated based on the expression of known marker genes.

In our case, however, we had no a priori knowledge of the exact number or types of cells that would result from the spontaneous differentiation of the EBs. Hence, we used three complementary approaches to annotate cells, capturing various perspectives on what might define a cell type in this data set. First, we identified cell types by clustering cells and annotating the cell types based on the genes that are highly expressed in each cluster. Second, we annotated cell types by considering the correlation of gene expression in our data with a reference data set of known primary cell types. For our third approach, we used a different perspective, and applied topic modeling to consider a less discrete definition of cell type.

For the first approach, we used a standard clustering analysis, the Louvain algorithm in Seurat, to identify groups of cells with similar transcriptomes (*Blondel et al., 2008*). To avoid making assumptions about the true number of cell types present, we repeated this analysis across different clustering resolutions (resolution 0.1, 0.5, 0.8, and 1). As expected, the number of clusters we identified varied greatly depending on the resolution (*Figure 1E–F*, *Figure 1—figure supplement 2*). We performed each subsequent analysis using clusters defined at multiple resolutions, to ensure that our qualitative conclusions are robust with respect to the number of clusters identified.

**Table 1.** Classification of Seurat cluster identity (clustering resolution 0.1) based on differential expression of marker genes.

| Cluster (Res. 0.1) | # cells in cluster | Top 10 marker genes by adj. P | Top 10 marker genes by logFC | Annotation |
|---|---|---|---|---|
| 0 | 17,693 | TERF1, PHC1, SEPHS1, UGP2, DPPA4, TBC1D23, JARID2, USO1, ZNF398, LRRC47 | DPPA5, DPPA3, GDF3, NANOG, FGF4, POU5F1, CBR3, PRDM14, DPPA2, TRIML2 | Pluripotent Cells |
| 1 | 14,383 | TPBG, FGFBP3, FZD3, LIX1, SDK2, BTBD17, DACH1, PLAGL1, DEK, ZNF219 | FEZF2, EMX2, LHX2, SOX3, PAX6, WNT7, BARX, SOX1, ZIC1, SIX3 | Early Ectoderm |
| 2 | 3086 | TNNI1, COL6A3, COL5A1, RGS4, ACTA2, TMEM88, DOK4, SLC40A1, HAND2, COL3A1 | RGS13, LUM, TECRL, DCN, HAND1, PITX1, COL3A1, SLN, IGF2, FIBIN | Mesoderm |
| 3 | 2673 | NR2F1, CNP, S100B, EDNRA, FGFBP3, ATP1A2, DNAJC1, ZEB2, PHACTR3, METRN | MPZ, PRSS56, ROPN1, SOX10, S100B, SCRG1, NPR3, MOXD1, TFAP2B, PHACTR3 | Neural Crest |
| 4 | 2368 | S100A16, LGALS3, GATA3, CST3, KRT19, FN1, EPSTI1, DYNLT3, HDHD3, PKP2 | APOA2, CST1, APOA1, APOC3, FGB, RBP4, S100A14, TTR, FGA, APOB | Endoderm |
| 5 | 1990 | TAGLN3, RTN1, NHLH1, STMN2, ELAVL2, FNDC5, PCBP4, ELAVL4, DCX, MLLT11 | NEUROD1, NHLH1, STMN2, NEUROD4, TBR1, STMN4, NEUROG1, SST, ELAVL3, SLC17A6 | Neurons |
| 6 | 295 | EGFL7, GNG11, RAMP2, IGFBP4, PPM1F, RASGRP3, RCSD1, MAP4K2, PLVAP, DOCK6 | PLVAP, CD34, CD93, CDH5, DIPK2B, PECAM1, EMCN, CRHBP, ESAM, ECSCR | Endothelial Cells |

For each clustering resolution, we calculated pseudobulk gene expression levels using cells from the same cluster, individual, and replicate. To identify marker genes expressed in each cluster, we used *Limma* and *voom* to perform differential expression analysis (Materials and methods) using the pseudobulk estimates. For example, considering the gene expression data of the seven clusters identified at resolution 0.1, we found that the most significantly upregulated genes in each cluster included known marker genes for pluripotent cells (cluster 0), early ectoderm (cluster 1), mesoderm (cluster 2), neural crest (cluster 3), endoderm (cluster 4), neurons (cluster 5), and endothelial cells (cluster 6) (*Figure 1E*, *Table 1*). Using this approach provides a confident set of broad cell type categories present in these data. At higher resolutions, DE analysis between clusters enabled annotation of some more specific cell types; for example, at clustering resolution 1, cluster 19 is characterized by higher expression of hepatocyte marker genes FGB, TTR, and AFP. More generally, however, confident cell type classification of Seurat clusters at higher resolution based on DE alone proved difficult (*Supplementary files 4-7*).

To pursue the second approach, we annotated cells by comparing our gene expression data to available reference sets of scRNA-seq data from primary fetal tissues, human embryonic stem cells (hESCs), and hESC-derived EBs (*Cao et al., 2020*; *Han et al., 2020*). To do this, we first integrated our data set with the reference data sets and visualized cells with UMAP (*Figure 2A*, *Figure 2—figure supplement 1*, Materials and Methods). We observed that reference hESCs cluster closely with pluripotent EB cells. We also observed that the hESC-derived EBs and our iPSC-derived EBs tend to occupy the same areas in UMAP space, implying high overall similarity in cell type composition despite differing protocols for EB differentiation (and despite the fact that the experiments were performed in different labs). EB cells also show overlap with many primary fetal cell types (*Figure 2B–C*, *Figure 2—figure supplement 2*). For example, EB cells annotated as neural crest based on our gene expression analysis, overlap with primary fetal cell types derived from neural crest, such as Schwann cells and enteric nervous system (ENS) glia (*Figure 2B*). EB cells annotated as neurons based on our gene expression analysis overlap with fetal neuronal subtypes, including inhibitory neurons, excitatory neurons, granule neurons, ENS neurons, and others. EB cells also show overlap with populations of cells that are rare in the fetal data set, including AFP_ALBpositive cells (hepatic cells), thymic epithelial cells, and lens fibre cells (*Figure 2C*).

Encouraged by these observations, we expanded the annotation of our EB cells (which up to this point were based on the expression of known marker genes) by using the known annotations of the reference primary fetal cell type data set. Specifically, we transferred cell annotations to EB cells based on the nearest reference cells in harmony-corrected PCA space (*Figure 2D*). Using this approach, we found EB cells representing 66 of the 77 primary cell types present in the reference fetal data set (*Supplementary file 1*). The most common annotation was hESC (80% of EB cells); this can be partially attributed to the high proportion of pluripotent cells in our EB data set, but also to the fact that the reference fetal data set does not include many early developmental cell types. Indeed, many cells annotated as hESC here are likely to represent immature, differentiating cells which are no longer pluripotent but whose transcriptional profiles more closely match hESCs than the more highly differentiated fetal cell types present in the reference data set. In this sense, EB data sets may capture transient developmental cell types that are difficult or impossible to study even in fetal primary samples. Outside the hESCs, many fetal cell types are only represented by small populations of EB cells. For example, only one EB cell is annotated as a thymocyte, and only one cell is annotated as a myeloid cell. These observations indicate that, in the future, we can benefit from a deeper sampling of EB single cells in order to properly explore their true cell type composition. Overall, annotation based on the reference set revealed the presence of dozens of diverse cell types in EBs.

## Differentiation efficiency across individuals

To assess the differentiation efficiency of each individual in each replicate, we calculated the proportion of cells assigned to each cluster as resolution 0.1 (*Figure 1G*). While EBs from two of the individuals in our study differentiated efficiently across all replicates, we observed that 89% of cells from individual 18858 were assigned to cluster 0, the cluster annotated as pluripotent cells based on differential expression of marker genes (*Table 1*, *Supplementary file 3*). The EBs from this line do differentiate, producing high quality cells assigned to clusters representing each germ layer (*Figure 1G*, *Figure 1—figure supplement 3*), but these EB have overall markedly lower differentiation efficiency

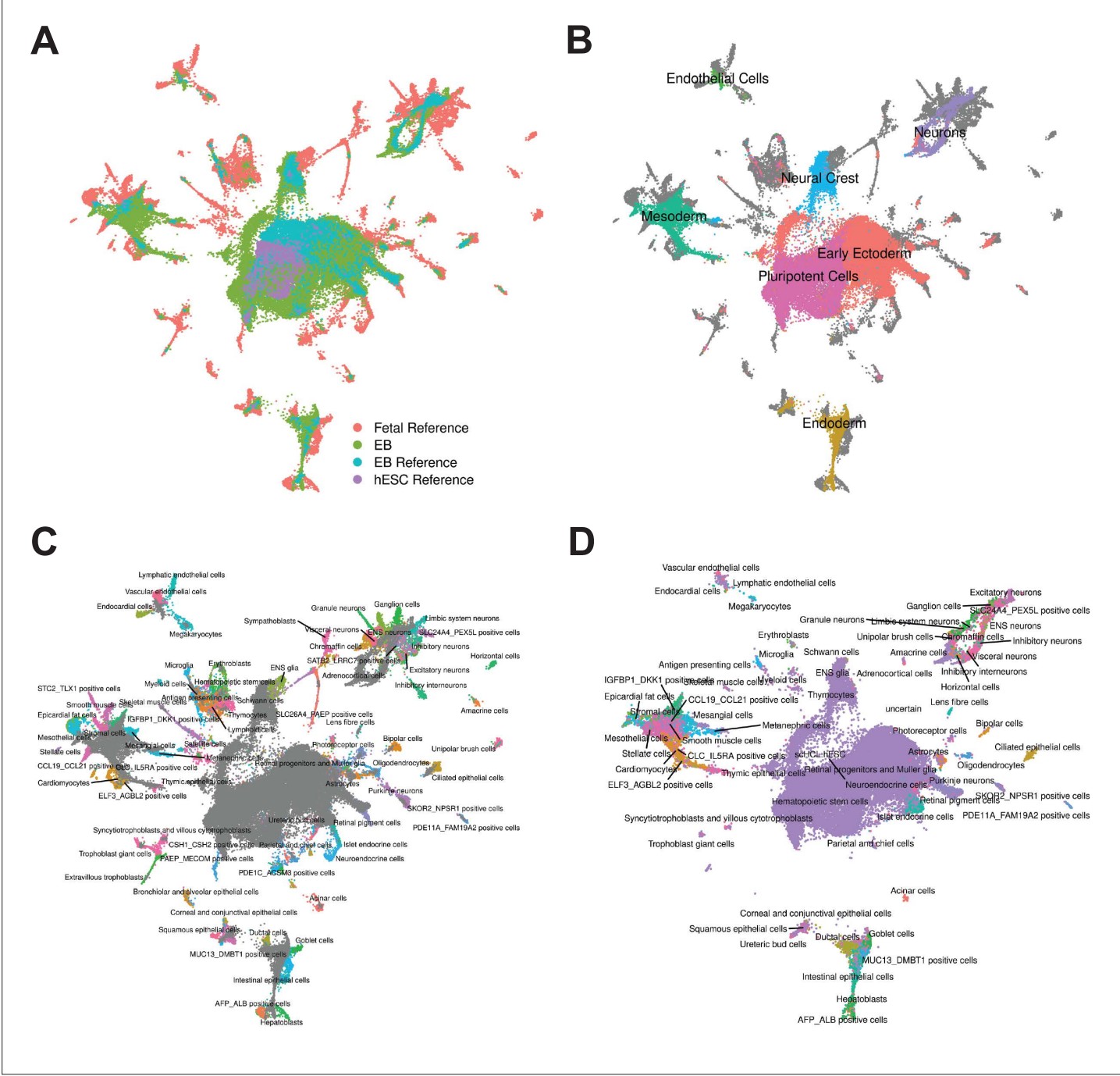

**Figure 2.** Reference Integration and cell type annotation with lines 18511, 18858, and 19160. (**A**) UMAP visualization of EB cells from this study and cells from reference data sets of fetal cell types, Day 20 EBs, and hESCs after integration. Cells are colored by data set. (**B**) UMAP visualization of EB cells from this study and cells from the fetal reference after integration. Cells are colored by Seurat cluster identity at clustering resolution 0.1, with gray points representing cells from the fetal reference set. (**C**) UMAP visualization of EB cells from this study and data from the fetal reference after integration. Cells are colored by cell types present in the fetal reference data set, with gray points representing EB cells. (**D**) UMAP visualization of EB cells from this data set with annotations transferred from the fetal and hESC reference sets.

The online version of this article includes the following figure supplement(s) for figure 2:

**Figure supplement 1.** UMAP visualization of EB cells from lines 18511, 18858, and 19160 and cells from each reference set after integration of separated data set.

**Figure supplement 2.** UMAP visualization of EB cells from lines 18511, 18858, and 19160 and fetal reference cells after integration.

**Figure supplement 3.** Differential expression of known marker genes in reference annotated EB cell types in cells from lines 18511, 18858, and 19160.

than EBs from individuals 18511 and 19160. To determine whether individual 18858 is an outlier, and more generally estimate how often EB differentiation is less efficient, we differentiated an additional five human iPSC lines from individuals 18856, 18912, 19140, 19159, and 19210 (Materials and methods). We were reassured to find that EBs from additional lines differentiated efficiently, with cell type composition similar to 18511 and 19160 (*Figure 1H*). These results suggest that poor differentiation efficiency is expected to be rare among the YRI iPSC panel. We further explored the robustness of cell type composition by integrating the additional lines with the fetal and hESC reference data sets using the same methodology as was used for the original lines (*Figure 3*, *Figure 3—figure supplement 1*, *Figure 3—figure supplement 2*). We find that annotations assigned to cells from these additional lines represented 66/77 fetal cell types; this set of annotations included several cell types that were missing in the original three lines, but also excludes several that were seen in the original three lines (*Supplementary file 2*). Again, we observed that most EB cells from the additional lines are annotated as hESC (82% of cells), although many no longer express pluripotency markers and do express markers of various germ layers as we previously observed. Together, these results support our conclusions that EBs contain many diverse cell types, many of which likely capture earlier stages of development than are captured in fetal data.

## Topic modeling of the single-cell gene expression data

Both of the approaches we described above (clustering, and comparison to a reference data set) assume that 'cell types' are discrete categories. Accordingly, each cell has a single true identity, and cell type categories are assumed to be homogeneous and static. This definition of a cell type is intuitive and makes it practical to consider results from single-cell analysis in the context of the wealth of knowledge previously gained from bulk assays. However, partitioning cells into discrete groups is unlikely to capture the full degree of heterogeneity in gene expression of single cells. A particular cluster or cell 'type' may collapse multiple cell states, obscuring functionally distinct subgroups such as cells in different stages of the cell cycle. This problem becomes more apparent in data sets that include differentiating cells, which are expected to show varying degrees of similarity to a terminal cell type. In an alternate paradigm, cell type can be viewed as continuous, with the expression profile of each cell representing grades of membership to multiple categories (*Dey et al., 2017*). One method used to capture cell identity in this paradigm is topic modeling, which learns major patterns in gene expression within the data set, or topics, and models each cell as a combination of these topics. We applied topic modeling using *fastTopics* at a range of topic resolutions, identifying 6, 10, 15, 25, and 30 topics in our data. Some topics correspond closely to Seurat clusters, loaded on cells of a given cluster but not on others. For example, in the k = 6 topic analysis, topic 1 is loaded exclusively on cells assigned to Seurat cluster 4 (cluster resolution 0.1) which we previously annotated as endoderm (*Figures 4A–D and 1E*, *Table 1*). Compared to other topics, topic 1 shows an increase in expression of *FN1* and *AFP*, which are known markers of hepatocytes (*Figure 4E*, *Table 2*). Seurat clustering at higher resolution (resolution 1) results in further categorical division of this large endoderm group of cells into definitive endoderm and hepatocytes (*Figure 1F*). Topic modeling revealed that these cells actually exhibit variable grades of membership in topic 1 (in k = 6 topic model); this gradient captures a temporal continuum of differentiation.

Certain topics are shared across cells assigned to different Seurat clusters (*Figure 4A*, *Figure 4—figure supplement 3*). For example, topic 6 from the k = 6 topic analysis is loaded across all Seurat clusters; compared to all other topics, topic 6 shows increased expression of many ribosomal genes, housekeeping genes (*GAPDH*), and genes coding for proteins involved in cellular metabolism (*LDHA*) (*Figure 4—figure supplements 1–3*). This indicates that topic six captures patterns of gene expression associated with cellular processes and functions that are not specific to a particular cell type. This again highlights an advantage of topic modeling, enabling us to explore variation in the representation of gene expression profiles associated with processes shared across many cell types, simultaneously with identifying cell-type-specific patterns.

## Biological and technical sources of variation

Once we functionally annotated EB cells using the three approaches discussed above, we sought to understand the consistency in cell type composition across individuals and between replicates. Here, 'replicate' corresponds to a batch of EB differentiations in which each cell line was differentiated,

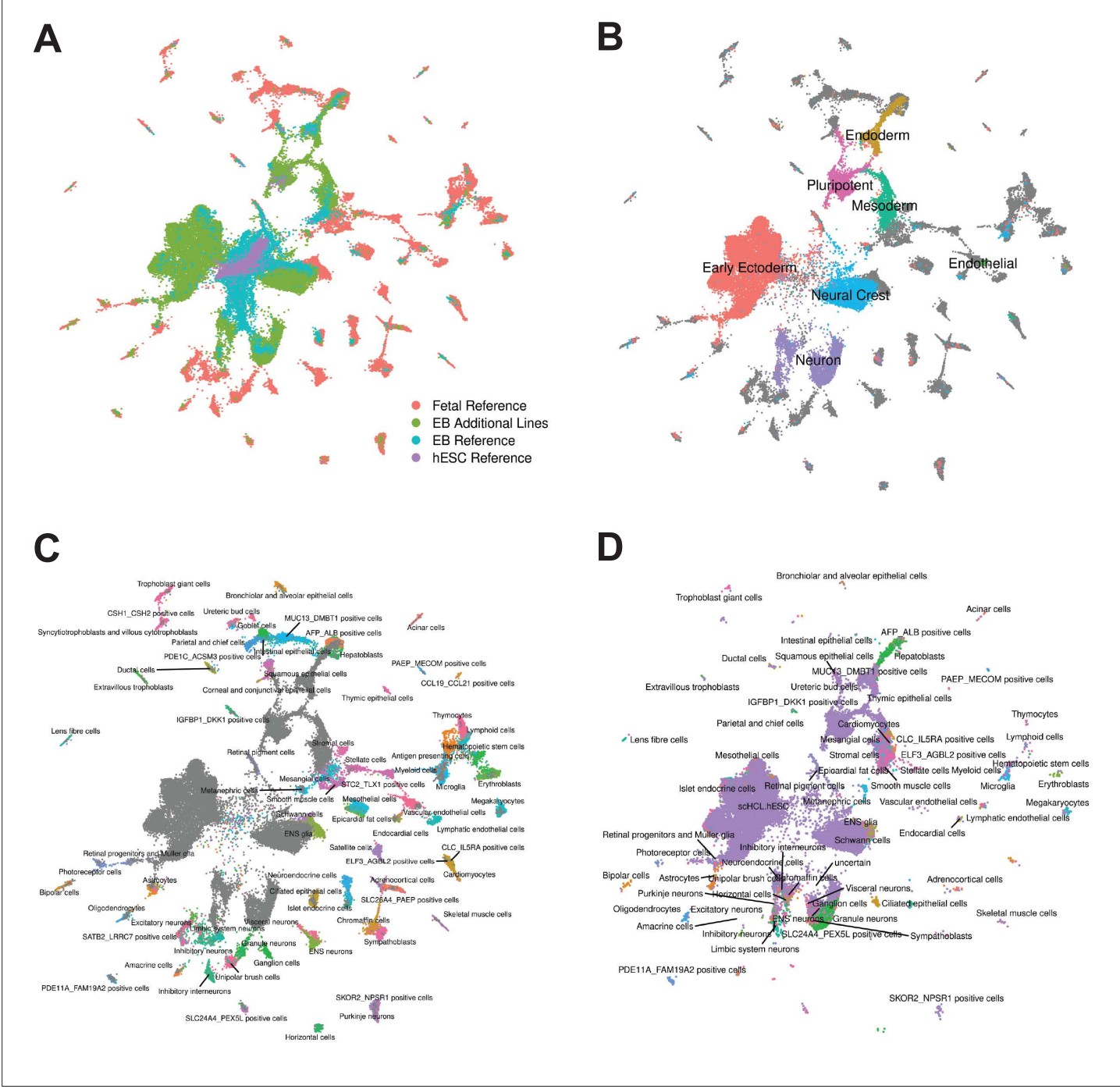

**Figure 3.** Reference Integration and cell type annotation with additional lines. (**A**) UMAP visualization of EB cells from this study and cells from reference data sets of fetal cell types, Day 20 EBs, and hESCs after integration. Cells are colored by data set. (**B**) UMAP visualization of EB cells from this study and cells from the fetal reference after integration. Cells are colored by broad cell type category assigned using clustering and marker gene expression, with gray points representing cells from the fetal reference set. (**C**) UMAP visualization of EB cells from this study and data from the fetal reference after integration. Cells are colored by cell types present in the fetal reference data set, with gray points representing EB cells. (**D**) UMAP visualization of EB cells from this data set with annotations transferred from the fetal and hESC reference sets.

The online version of this article includes the following figure supplement(s) for figure 3:

**Figure supplement 1.** UMAP visualization of EB cells from lines five additional YRI lines and cells from each reference set after integration of separated data set.

**Figure supplement 2.** UMAP visualization of EB cells from five additional YRI lines and fetal reference cells after integration.

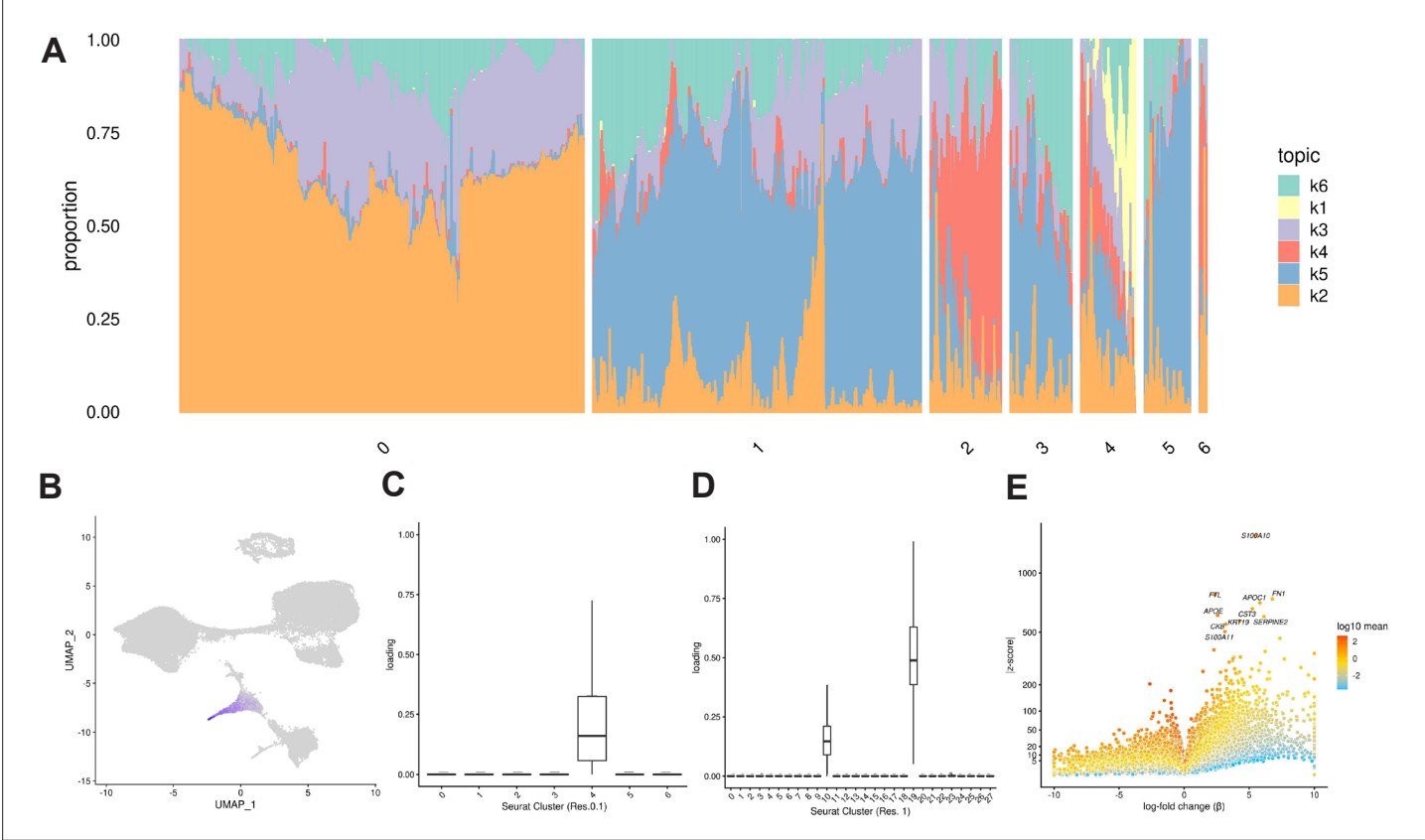

**Figure 4.** Topic modeling of EB cells. (**A**) Structure plot showing the results of topic modeling at k = 6. Plot includes a random subset of 5,000 EB cells divided by Seurat cluster at resolution 0.1. (**B**) UMAP projection of cells colored by loading of topic 1. (**C**) Box plot showing the loading of topic 1 from the k = 6 topic analysis on each Seurat cluster at clustering resolution 0.1. (**D**) Box plot showing the loading of topic 1 from the k = 6 topic analysis on each Seurat cluster at clustering resolution 1. (**E**) Volcano plot showing genes differentially expressed between topic 1 and all other topics from the k = 6 topic analysis. Points are colored by the average count on the logarithmic scale.

The online version of this article includes the following figure supplement(s) for figure 4:

**Figure supplement 1.** UMAP visualization of k = 6 topic loadings.

**Figure supplement 2.** Volcano plot showing genes differentially expressed in each topic from the k = 6 topic analysis.

**Figure supplement 3.** Topic loadings on Seurat clusters across clustering resolutions.

dissociated, and collected in tandem; 'replicate' therefore captures technical variation related to differentiation batch, dissociation batch, and single-cell collection batch. We began by calculating the proportion of cells that were assigned to each Seurat cluster at resolution 0.1 for each replicate. We then performed hierarchical clustering of the samples based on the proportion of cells in each Seurat cluster (*Figure 5—figure supplement 1*). Using this approach, replicate-individual samples cluster first by individual, indicating that cell type composition is distinct between individuals and is consistent between replicates of each individual. We repeated this analysis at a range of cluster resolutions and determined that this finding is robust with respect to the number of clusters (*Figure 5—figure supplement 1*).

We also repeated this analysis using topic loadings as a measure of cell type composition. We calculated the loading of each topic on each individual-replicate group and performed hierarchical clustering (*Figure 5—figure supplement 2*). Again, we found that at varying values of k, samples generally cluster by individual, but using the higher resolution topic-based approach, we also observed substantial variation between replicates (*Figure 5—figure supplement 2*). Individual 18858 always clusters away from the other two lines, due to the consistent and distinct distribution of cell types caused by low differentiation efficiency.

**Table 2.** Top 15 driver genes of each topic from the k = 6 topic model based on Z-score.

| Topic | Top 15 driver genes |
| --- | --- |
| k1 | S100A10, FTL, FN1, APOC1, CST3, APOE, SERPINE2, KRT19, CKB, S100A11, LGALS3, TMSB10, S100A16, AFP, PTGR1 |
| k2 | MT-CO2, MT-CO3, MT-CO1, MT-CYB, PRDX1, MT-ND4, MT-ATP6, GSTP1, MT-ND1, RPL8, APOE, RPSA, RPL12, PFN1, HMGA1 |
| k3 | PTMA, NCL, RPL23, SET, HSP90AB1, TPL27A, MT-ND4, L1TD1, SERBP1, TERF1, HSPD1, CENPF, DPPA4, MT-ATP6, UGP2 |
| k4 | S100A10, KRT19, S100A11, VIM, MDK, TMSB10, KRT8, SPARC, COL1A1, FN1, COL1A2, COL6A2, KRT18, TPM1, ANXA2 |
| k5 | TUBA1A, VIM, MARCKSL1, MARCKS, TUBA1B, MAP1B, ID3, CRABP1, PTMS, TMSB10, H1FX, STMN1, CENPV, CRABP2, NUCKS1 |
| k6 | RPS27, VIM, LDHA, GAPDH, IGFBP2, TUBA1A, APOA1, RPL13, TMSB10, S100A10, RPL6, RPL30, RPL9, RPS19, RPL37 |

We further characterized determinants of variation in our system by considering factors that contribute to variation in gene expression levels. Hierarchical clustering of pseudobulk expression estimates of cells from the same Seurat cluster (res. 0.1), replicate, and individual shows that, as might be expected, samples tend to cluster first by cell type (Seurat cluster), then by individual and replicate (*Figure 5A*). We performed variance partitioning using pseudobulk expression levels to estimate the relative contribution of cell type, individual, and replicate to overall patterns of gene expression variation (*Figure 5B*; *Hoffman and Schadt, 2016*). We found that replicate and individual explained approximately equal proportions of the variance (each explains a median value of ~5% of variance). Cell type identity explained the largest proportion of variation at all clustering resolutions tested (variance explained median value ~60% at clustering resolution 0.1), although this figure is likely exaggerated since cell type identity is determined by clustering, which will inherently maximize variation between cell types. Depending on clustering resolution, a median value of approximately 20–30% of variance is explained by residuals, which can be attributed to noise or other technical variation not specifically modeled (*Figure 5—figure supplement 3*). We then partitioned the variance in gene expression at single cell resolution (instead of using pseudobulk estimates) and found that replicate explains more variation on average than individuals, with cell type identity continuing to explain more variance than either (*Figure 5C*). At single-cell resolution, residuals explain a median value of 93% of the variation, which is expected due to the high degree of variance (both biological and technical) in gene expression profiles among individual cells.

To determine whether biological and technical factors contributed differently to variation between cell types, we also partitioned the variance due to replicate and individual in each Seurat cluster separately (*Figure 5—figure supplements 4–5*). The results are not uniform across clusters. At clustering resolution 0.1, individual contributes more to variation, on average, in clusters 0, 1, 4, and 5, while replicate contributes more to variation in clusters 2, 3, and 6. Notably, clusters 2, 3, and 6 include only a few cells from individual 18858 (*Supplementary file 3*). Studies that incorporate a larger number of cells will increase representation of rare cell types, which will increase power to study patterns of gene regulation. In every cluster, variation due to replicate dominates the variation of certain genes but not others. This complex structure indicates that, unlike most other eQTL studies – where adding individuals is always preferable to adding technical replicates – future studies of EBs need to implement study designs with multiple replicates to appropriately account for this variation.

Because individual variation contributes to overall patterns of variation in gene expression, EBs have the potential to be a powerful model to understand inter-individual variation in gene regulation across cell types and to map dynamic eQTLs. We performed a power analysis to better understand the relationship between power, sample size, and the total number of individual cells analyzed, or the experiment size (*Figure 6*). Assuming a simple linear regression to map eQTLs and a conservative

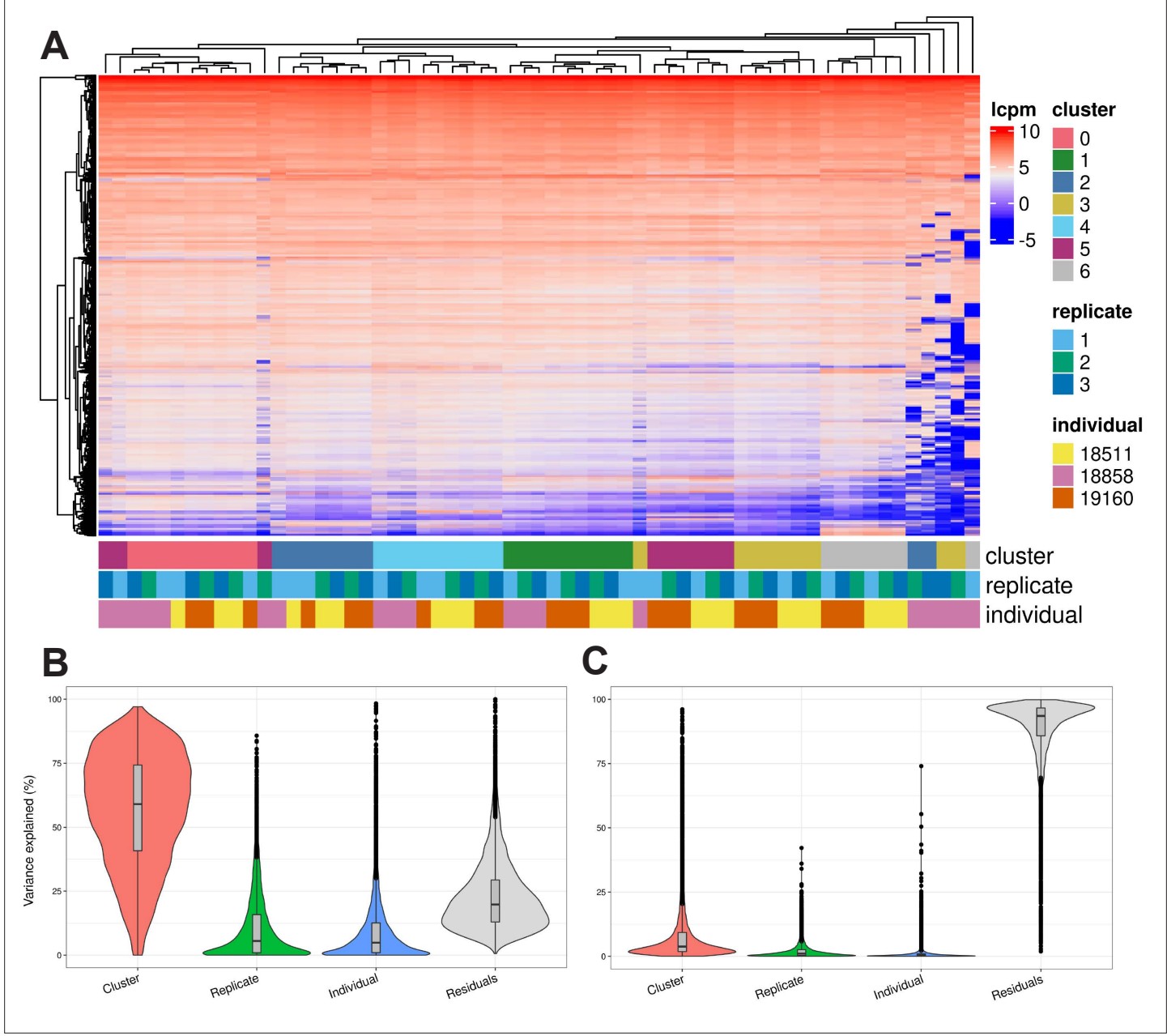

**Figure 5.** Exploration of the biological and technical variation in gene expression across EB cells. (**A**) Heatmap showing hierarchical clustering of cells based on normalized gene expression. This analysis uses only genes expressed in at least 20% of cells in at least one cluster (at clustering resolution 0.1) and does not include ribosomal genes. (**B**) Violin plot showing the percent of variance in gene expression explained by cluster (resolution 0.1), replicate, and individual in this data set after partitioning variance in pseudobulk samples. (**C**) Violin plot showing the percent of variance in gene expression explained by cluster (resolution 0.1), replicate, and individual in this data set after partitioning variance at single-cell resolution.

The online version of this article includes the following figure supplement(s) for figure 5:

**Figure supplement 1.** Hierarchical clustering of samples' individual-replicate groups by the proportions of cells from each group assigned to each Seurat cluster across clustering resolutions.

**Figure supplement 2.** Hierarchical clustering of samples' individual-replicate groups by the loading of each topic with k = 6, k = 10, k = 15, k = 25, and k = 30 topics.

**Figure supplement 3.** Variance explained by biological and technical factors at higher clustering resolutions.

**Figure supplement 4.** Variance partitioning by Seurat cluster using pseudobulk samples.

**Figure supplement 5.** Median percent of variance explained by replicate and individual in each cluster using pseudobulk samples.

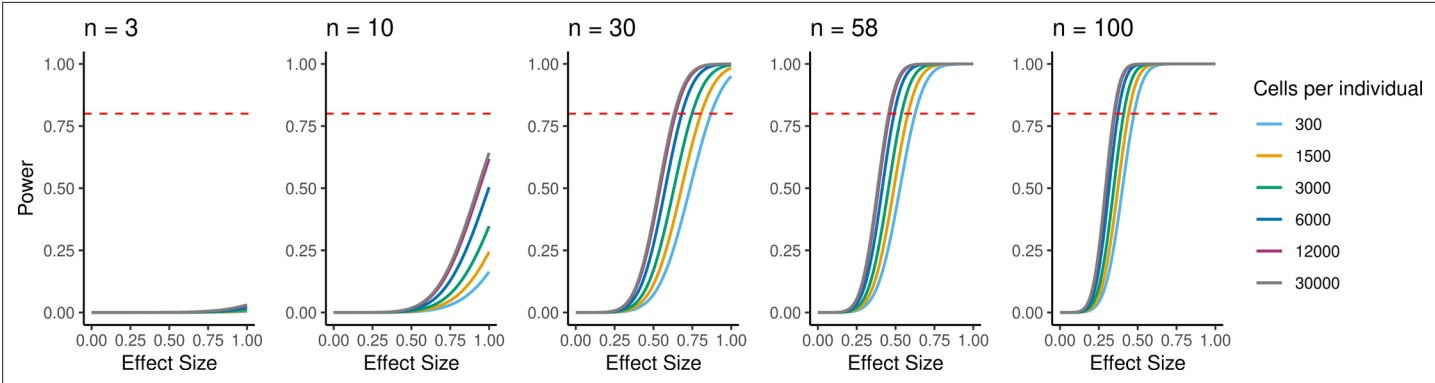

**Figure 6.** Power to detect eQTLs. Power is a function of effect size, sample size, experiment size, and significance level. Power curves are computed for a range of sample sizes and experiment sizes (cells per individual). The horizontal red line represents a power to detect eQTLs of 0.80.

Bonferroni correction for multiple testing (FWER = 0.05, Materials and methods), we estimated that an experiment consisting of 58 individuals with 3000 cells collected per individual, collected across three replicates (experiment size of 174,000 cells total), would provide 93% power to detect eQTLs with a standardized effect size of 0.6. These assumptions represent an experimentally tractable study design, and a conservative estimate of standard and dynamic eQTL effect sizes, suggesting this could be a powerful system for QTL studies across diverse human cell types.

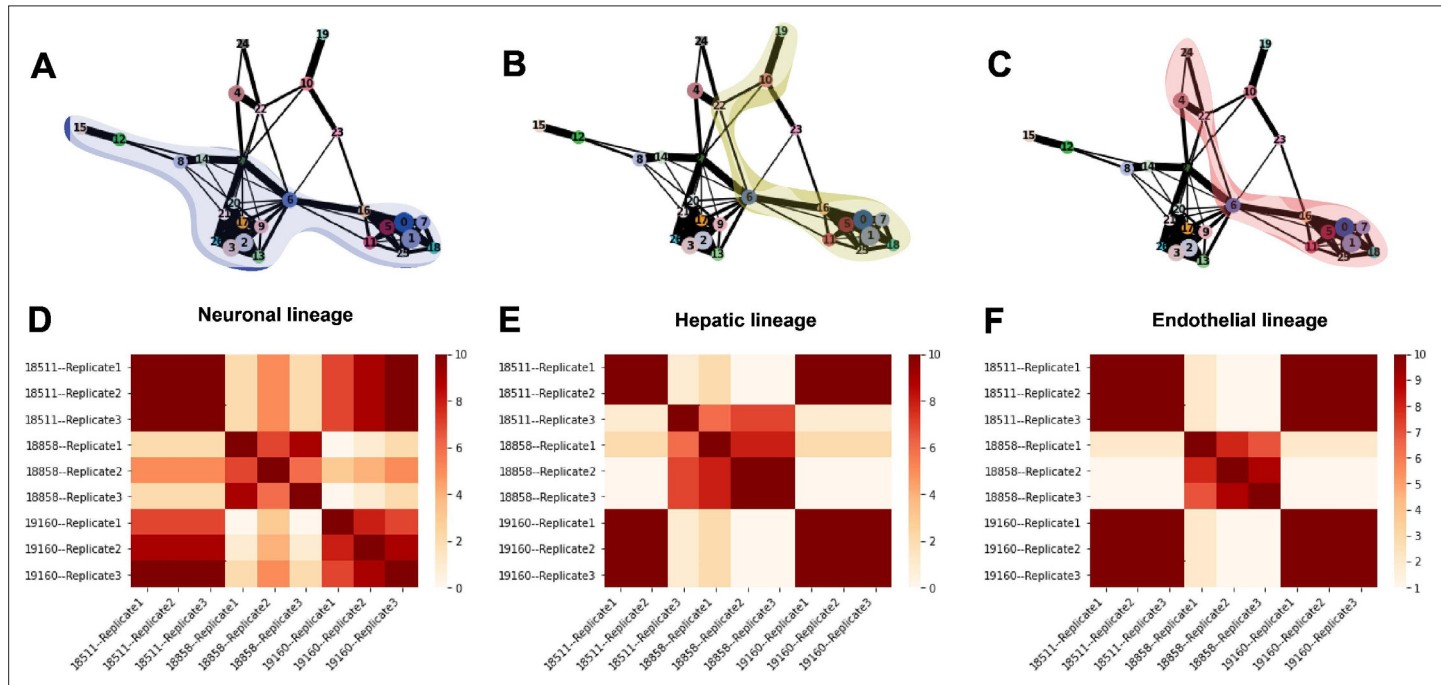

**Figure 7.** Trajectory inference and identification of dynamic gene modules. (**A–C**) PAGA graphs highlighting the neuronal lineage (**A**), the hepatic lineage (**B**), and the endothelial lineage (**C**). Nodes are defined by Seurat clusters at resolution 1. (**D–F**) Heatmaps showing the frequency with which individual-replicate groups were assigned to the same cluster after running split-GPM 10 times in the neuronal, hepatic, and endothelial lineages.

The online version of this article includes the following figure supplement(s) for figure 7:

**Figure supplement 1.** Trajectory inference with PAGA.

**Figure supplement 2.** Marker gene expression in Seurat clusters aids tracing of developmental lineages.

**Figure supplement 3.** Cluster assignment by Split-GPM and gene set enrichment in the neuronal lineage.

**Figure supplement 4.** Cluster assignment by Split-GPM and gene set enrichment in the hepatic lineage.

**Figure supplement 5.** Cluster assignment by Split-GPM and gene set enrichment in the endothelial lineage.

## Dynamic patterns of gene expression

Arguably, the most attractive property of single-cell data from the EB system is the ability to study dynamic gene regulatory patterns throughout differentiation. In order to explore dynamic patterns of gene expression, we inferred developmental trajectories using PAGA (*Wolf et al., 2019*). The PAGA graph shows edges that represent likely connections between cell clusters (clustering resolution 1) and we were able to trace developmental trajectories through these paths (*Figure 7A–C*, *Figure 7—figure supplement 1A-C*). Since the EBs still include undifferentiated pluripotent cells, we were able to define rooted trajectories to each germ layer beginning at the known starting point. Trajectories toward endoderm and mesoderm proceed through cluster 22, which expresses primitive streak marker *MIXL1*, showing recapitulation of developmental trajectories defined during gastrulation (*Figure 1F*, *Figure 7—figure supplement 1*, *Figure 7—figure supplement 2*). Hepatocytes (cluster 19), an endoderm-derived cell type, branch off of the endoderm cluster (cluster 10) (*Figure 7B*, *Figure 7—figure supplement 2*). Endothelial cells (cluster 24), which are derived from mesoderm, branch off from the mesoderm cluster (cluster 4) (*Figure 7C*, *Figure 7—figure supplement 2*), and neurons (clusters 12, 15), an ectoderm-derived cell type, branch off from the early ectoderm clusters (clusters 2, 3, 8, 9, 13, 14, 17, 20, 21, 26, and 27) (*Figures 7A and 1F*, *Table 1*, *Figure 7—figure supplement 2*).

We then assigned pseudo-time values to each cell using the diffusion pseudo-time method with pluripotent cells (cluster 1) defined as the root (*Haghverdi et al., 2016*; *Figure 7—figure supplement 1D*, *Figure 1F*). We manually traced high confidence trajectories through the data representing the progression from pluripotent cells to hepatocytes (clusters 0, 1, 5, 6, 7, 10, 11, 16, 18, 19, 25, and 22) (*Figure 7B*), pluripotent cells to endothelial cells (clusters 0, 1, 4, 5, 6, 7, 11, 16, 18, 22, 24, and 25) (*Figure 7C*), and pluripotent cells to neurons (clusters 0, 1, 2, 3, 5, 6, 7, 8, 9, 11, 12, 13, 14, 15, 16, 17, 18, 20, 21, 26, and 27) (*Figure 7A*). For groups of clusters with a higher degree of connectivity (e.g. clusters expressing pluripotent markers and clusters expressing early ectoderm markers), all clusters within the region with high connectivity were included in the trajectory to avoid choosing an arbitrary path through these clusters. Next, we applied split-GPM, an unsupervised probabilistic model, to infer dynamic patterns of gene expression within a particular developmental trajectory, while simultaneously performing clustering of genes and samples. Split-GPM is built for use with time course, bulk RNA-seq data; therefore, we calculated pseudobulk expression values for individual-replicate groups within decile bins of pseudo-time. We were able to identify gene modules with distinct dynamic patterns of expression along the trajectories to neurons, hepatocytes, and endothelial cells (*Figure 7—figure supplements 3–5*).

Gene set enrichment analysis of these modules shows expected dynamic patterns (*Figure 7—figure supplements 3–5*). For example, we found that gene modules that increase expression through pseudo-time along the differentiation trajectory to hepatocytes, which are the predominant cell type of the liver and are responsible for the production of bile, are enriched for the hallmark bile acid metabolism and fatty acid metabolism gene sets. In the trajectory leading to endothelial cells, which are derived from mesoderm, we found that a gene module with high expression at intermediate pseudo-time values is enriched for hallmark genes expressed during the epithelial-mesenchymal transition, which is essential for mesoderm formation (*Evseenko et al., 2010*). In all three trajectories, gene modules characterized with higher expression at low pseudo-time values show enrichment for gene sets related to the cell cycle; this is expected because pluripotent cells at the lowest pseudo-time values tend to grow and divide faster than more differentiated and more mature cell types, which often exit the cell cycle (*Buttitta and Edgar, 2007*).

To determine the consistency in dynamic patterns of gene expression between replicates and individuals, we ran split-GPM ten times on cells from the neuron, hepatocyte, and endothelial cell lineages and observed how often each pair of individual-replicate samples clustered together (*Figure 7D–F*). In the neuronal and endothelial lineages, all three replicates of 18511 always clustered together and often cluster with replicates of 19160, indicating that these two lines share similar expression dynamics in these trajectories (*Figure 7D and F*). Replicates of 18858 often clustered together and rarely clustered with the other individuals, suggesting that not only did 18858 have poor differentiation efficiency, but cells that did differentiate show a distinct pattern of expression dynamics. In the hepatocyte lineage, we observed stronger replicate-specific differences (*Figure 7E*). Replicates of individual 19160 still tended to cluster together and to cluster with replicates 1 and 2 of 18511.

Replicate 3 of 18511 rarely clustered with the other replicates of that individual, indicating that there were replicate-specific effects on dynamic gene expression.

## Discussion

This work represents a thorough exploration of heterogeneity in single cell data obtained from human EBs towards the goal of establishing this system as a tool to enable studies of variation in human gene regulation across a range of spatially and temporally diverse cell types. We used iPSC-derived EBs because this in vitro model system circumvents the logistical challenges and ethical barriers associated with studies of primary human developmental tissues. This system has key advantages over studies of primary tissues; for example, we are able to control cellular environment in vitro and intentionally design experiments with respect to biological factors including age, sex, and ancestry. Further, we can generate EBs comprised of the same set of diverse cell types from large samples of individuals, enabling high-powered comparisons of cell-type-specific gene expression.

In subsequent studies, we plan to leverage EBs to identify QTLs and dynamic QTLs across diverse terminal and differentiating cell types. This, of course, raises an ostensibly critical question: to what extent do the cell types derived from EBs faithfully model immature, developing cells in vivo? There is no doubt that the in vitro EB differentiated cells are not a perfect model of primary cell types. The question is whether EB cells are sufficiently representative of primary cell types to be informative. To address this question, we performed several analyses, which suggest that the EB model can be useful. Specifically, we found that EB cells express known cell-type-specific marker genes, including markers of known developmental stages. EB cells also cluster with more than 60 diverse primary cell types from a reference panel of fetal tissues and hESCs, including rare fetal cell types. Lastly, we identified gene modules with dynamic expression patterns that match broad expectations of developmental biology. Together, these results provide evidence that EBs are a suitable model of both terminal and developmental cell types.

Moreover, EBs may be a useful model for understanding the genetic underpinnings of human traits and diseases regardless of the degree to which they faithfully model human development. EB-derived cells represent a wealth of previously unstudied cell states and dynamic processes. Hypothetically, QTLs identified in these cell types still represent biologically meaningful differences in genetic control of gene regulation, whether they manifest in human development or in adult tissues upon a particular environmental exposure. To provide an anecdotal example of this reasoning, we considered previously collected data from an in vitro differentiation experiment. We took a closer look at the 28 middle-dynamic eQTLs Strober et al. identified during the differentiation of iPSCs to cardiomyocytes (*Strober et al., 2019*). Middle-dynamic eQTLs have their strongest genetic effects at intermediate stages of the differentiation time course, and most of them (25/28) were identified exclusively at these intermediate stages of differentiation. Accordingly, these eQTLs are active in early in vitro differentiating cells whose fidelity to primary developing cell types has not been ascertained. These 28 dynamic eQTLs were entirely novel and had not been identified as *cis* eQTLs in any tissue in the GTEx data set. Strober et al. reported that one of these middle-dynamic eQTLs was also found to overlap a GWAS variant associated with body mass index and red blood cell count. This finding highlights that dynamic eQTLs acting in early cell types in in vitro differentiations may affect long-term disease risk in adults.

To further explore the utility of dynamic eQTLs identified in in vitro differentiations, we used GTEx data to ask whether the middle-dynamic eQTLs are associated with inter-individual variation in trans gene expression or cell composition, either of which could indicate lasting downstream effects in adult tissue from transient dynamic *cis* eQTLs. *Trans* eQTL associations are more tissue-specific than *cis* eQTLs, but *trans* eQTLs are much harder to identify because of their small effect sizes and the requirement to test the association of every locus with every gene. Here, we identified a middle dynamic eQTL SNP (rs6700162) from Strober et al. that, in GTEx data, is associated with fibroblast cell type proportions in HLV (heart left ventricle; $p < 0.0009$) and with cardiac muscle cell proportions in HLV ($p < 0.003$). This SNP was also found to have a *trans* eQTL p-value of $1 \times 10^{-5}$ in coronary artery. Without the prior knowledge provided by dynamic eQTL data from the in vitro differentiated cardiomyocytes, it would have been impossible to identify these associations using adult primary tissues because the burden of multiple testing within the entire GTEx data set is considered is prohibitively large. This example implies that developing EB cells could be used to understand how transient effects on gene expression are propagated into functional, long-lasting consequences downstream.

In summary**,** human EBs have the potential to be a powerful system for the identification of dynamic eQTLs. In this pilot study, we performed foundational analyses to better understand how to appropriately conceptualize heterogeneity in this kind of data and how to best design large-scale studies of EBs. We explored cell type composition of EBs in three paradigms; first, with discrete cell types identified with a traditional clustering algorithm, then with more continuous cell "types" identified with topic modeling, and finally exploring dynamic changes in gene expression along trajectories using pseudotime. Cell types defined by discrete clustering are often easier to interpret because they can be contextualized with marker genes and reference data sets defined with bulk sequencing. We conclude, however, that topic modeling is more appropriate for highly heterogeneous single cell data sets like this one. We also explored sources of variation in cell type composition and gene expression. We found that individual variation primarily contributes to patterns in cell type composition based on both discrete clustering and topic modeling. However, variation between replicates is non-negligible, indicating that future studies should focus on inter-individual variation in cell type composition. We also found that technical variation between replicates contributes to variation in gene expression. Future efforts to map regulatory QTLs in EBs should implement study designs with multiple replicates to appropriately correct for batch effects. Overall, this pilot study has laid the groundwork to transform EBs into a powerful model system for the understanding of human gene regulation.

# Materials and methods

**Key resources table**

| Reagent type (species) or resource | Designation | Source or reference | Identifiers | Additional information |
|---|---|---|---|---|
| Cell line (*Homo-sapiens*) | 18511 | PMID:29208628 | | |
| Cell line (*Homo-sapiens*) | 19160 | PMID:29208629 | | |
| Cell line (*Homo-sapiens*) | 18858 | PMID:29208630 | | |
| Cell line (*Homo-sapiens*) | 18856 | PMID:29208631 | | |
| Cell line (*Homo-sapiens*) | 18912 | PMID:29208632 | | |
| Cell line (*Homo-sapiens*) | 19140 | PMID:29208633 | | |
| Cell line (*Homo-sapiens*) | 19159 | PMID:29208634 | | |
| Cell line (*Homo-sapiens*) | 19210 | PMID:29208635 | | |

## Samples

We used iPSC lines from eight unrelated individuals from the Yoruba HapMap population to form EBs. The iPSC lines were reprogrammed from lymphoblastoid cell lines (LCLs) and were characterized and validated previously (*Banovich et al., 2018*). The original LCL lines were genotyped by the HapMap project and identity of the stocks used in this study is confirmed by scRNA-seq data collected for this study (*Belmont et al., 2003*). All cell lines used in this study tested negative for mycoplasm. Lines 18511, 18858, 18912, 19140, and 19159 are from female individuals. Lines 19160, 18856, and 19210 are from male individuals. Preprocessing and analysis of lines 18511, 18858, and 19160 are described throughout the Materials and methods section. Preprocessing and analysis of lines 18856, 18912, 19140, 19159, and 19210 is restricted to the Methods section titled 'Assessment of cell type composition and differentiation efficiency in five additional lines'.

## iPSC maintenance

We maintained feeder-free iPSC cultures on Matrigel Growth Factor Reduced Matrix (CB-40230, Thermo Fisher Scientific) with StemFlex Medium (A3349401, Thermo Fisher Scientific) and Penicillin/ Streptomycin (30,002 Cl, Corning). We grew cells in an incubator at 37 °C, 5% $CO_2$, and atmospheric O2. Every 3–5 days thereafter, we passaged cells to a new dish using a dissociation reagent (0.5 mM EDTA, 300 mM NaCl in PBS) and seeded cells with ROCK inhibitor Y-27632 (ab120129, Abcam).

## EB formation and maintenance

We formed EBs using a modified version of the STEMCELL Aggrewell400 protocol. Briefly, we coated wells of an Aggrewell 400 24-well plate (34415, STEMCELL) with anti-adherence rinsing solution (07010,

STEMCELL). We dissociated iPSCs and seeded them into the Aggrewell400 24-well plate at a density of 1,000 cells per microwell ($1.2 \times 10^6$ cells per well) in Aggrewell EB Formation Medium (05893, STEM-CELL). After 24 hr, we replaced half of the spent media with fresh Aggrewell EB Formation Medium. Forty-eight hr after seeding the Aggrewell plate, we harvested EBs and moved them to an ultra-low attachment six-well plate (CLS3471-24EA, Sigma) in E6 media (A1516401, ThermoFisher Scientific). We maintained EBs in culture for an additional 19 days, replacing media with fresh E6 every 48 hr. We performed three replicates of EB formation on different days; each replicate included all three lines.

## EB Dissociation

We collected and dissociated EBs 21 days after formation. We dissociated EBs by washing them with phosphate-buffered saline (PBS) (Corning 21–040-CV), treating them with AccuMax (STEMCELL 7921) and incubating them at 37 °C in for 15–35 min. After 10 min in Accumax, we pipetted EBs up and down with a clipped p1000 pipette tip for 30 s. We repeated pipetting every 5 min until EBs were completely dissociated. We then stopped dissociation by adding E6 media and straining cells through a 40 µm strainer (Fisherbrand 22-363-547). We resuspended cells in PBS and counted them with a TC20 Automated Cell Counter (450102, BioRad). Before scRNA-seq, we mixed together an equal number of cells from each line.

## Single-cell sequencing

We collected scRNA-seq data using the 10x Genomics V3.0 kit. Single-cell collections for this experiment were mixed with cells from a larger experiment in all three replicates. From the first replicate of EB differentiations, we mixed EB cells YRI individuals 18511, 18858, and 19160 with EB cells from an additional three humans and chimpanzees (nine individuals total). Even numbers of cells from all nine individuals were collected across nine lanes of a 10x chip, targeting 10,000 cells per lane. In this replicate, reagents from three different 10x kits were used. From replicates 2 and 3 of EB differentiation, EBs were only generated from the same three YRI individuals (18511, 18858, and 19160) and the three chimpanzees (six individuals total). In each replicate, we mixed even numbers of cells of each individual and collected cells in four lanes of a 10x chip, targeting 10,000 cells per lane, and samples were processed using reagents from a single 10x kit.

Libraries were sequenced using paired-end 100 base pair sequencing on the HiSeq 4000 in the University of Chicago Functional Genomics Core. For libraries from replicate 1, we mixed equal proportions of each of the six EB libraries and sequenced the pooled samples on one lane of the HiSeq 4000. Preliminary analyses showed that two of these lines were low quality. We remade one of the low-quality libraries and discarded the other. We then mixed equal proportions of the remade library with the remaining three libraries from replicate one and sequenced the pooled samples on one lane of the HiSeq 4000. Preliminary analyses indicated that three out of four of these libraries were below optimal quality, but would produce usable data. We then pooled together samples from the final eight libraries from replicate 1, mixing equal parts of each of the five high-quality libraries with half the amount of the other three, and deep-sequenced this pool on eight lanes of the HiSeq 4000. For replicate two libraries, we mixed equal parts of all four libraries and sequenced them on one lane. After confirming that each library was high-quality, we deep-sequenced the same pool on six additional lanes of the HiSeq 4000. For replicate three libraries, we mixed equal parts of all four libraries and sequenced them on one lane. After confirming that each library was high-quality, we deep-sequenced the same pool on four additional lanes of the HiSeq 4000. In all cases, the number of lanes for deep sequencing was calculated to reach 50% saturation.

## Alignment and sample deconvolution

We used *STARsolo* to align samples to both the human genome (GRCh38) (*Dobin et al., 2013*) and the chimpanzee genome (January 2018; Clint_PTRv2/panTro6). We used gene annotations from ensembl98 and transmapV5, respectively. In order to separate human cells from chimpanzee cells, we identified discordant reads – those that mapped with different scores in each species. We identified a cell as human if (1) at least five discordant reads that had a higher human mapping score and (2) at least 80% of discordant reads had a higher human mapping score. The remainder of analyses in this work were restricted to these human cells. We demultiplexed individual samples and identified doublets using *demuxlet* (*Kang et al., 2018*). For this demultiplexing with *demuxlet*, we used

previously collected and imputed genotype data for these three Yoruba individuals from the HapMap and 1000 Genomes Project (*Auton et al., 2015*; *Belmont et al., 2003*).

## Filtering and integration

We ran *EmptyDrops* to identify barcodes tagging empty droplets and kept only barcodes with a high probability of tagging a cell-containing droplet (i.e. we kept cells with an *EmptyDrops* FDR < 0.0001) (*Lun et al., 2019*). We removed cells labeled as doublets or ambiguous by *demuxlet*, keeping only barcodes labeled as singlets. We also filtered the data to include only high-quality cells expressing between 3% and 20% mitochondrial reads and expressing more than 1500 genes. We normalized data from each 10x lane using *SCTransform* in Seurat (*Butler et al., 2018*; *Hafemeister and Satija, 2019*). In total, we obtained 42,488 high-quality cells. We then merged data from each of the 10x lanes from all replicates, scaled the data, and ran principal components analysis (PCA) using 5000 variable features. We then integrated data with Harmony to correct the PCA embeddings for batch effects and individual effects (*Korsunsky et al., 2019*).

## Clustering and cell type annotation

To cluster the data, we applied Seurat's *FindNeighbors* using 100 dimensions from the Harmony-corrected reduced dimensions, followed by *FindClusters* at resolutions 0.1, 0.5, 0.8, and 1.

We performed differential expression analysis using the *limma* R package (*Ritchie et al., 2015*). First, we filtered genes to include only those expressed in at least 20% of cells in at least one cluster at a given clustering resolution. We then calculated pseudobulk expression values for each individual-replicate-cluster grouping (i.e. cells from the same individual, same replicate, and same cluster assignment). Accordingly, we define pseudobulk expression values as the sum of normalized counts in each group. Next we TMM-normalized pseudobulk expression values and used *voom* to calculate a weighted gene expression value to account for the mean-variance relationship (*Law et al., 2014*). We then fit the following linear model:

$$Y = 0 + \beta_{cluster} * x + \beta_{replicate} * x + \beta_{individual} * x$$

We used contrasts to first test for differential expression of each cluster compared to all other clusters and then to test for differential expression between pairs of similar clusters to find distinguishing markers. We annotated cell type identity of each cluster based on significant differential expression of the known marker genes.

## Assessment of cell type composition and differentiation efficiency in five additional lines

To evaluate the cell type composition resulting from EB differentiation of YRI iPSC lines more generally, we differentiated five additional randomly chosen lines (18856, 18912, 19140, 19159, and 19210) from the YRI iPSC panel. We differentiated and dissociated iPSCs in parallel using the same protocols described above. After dissociation, we mixed cells from each individual in equal proportions and collected scRNA-seq data using the 10x genomics V3.1 kit, targeting collection of 10,000 cells per lane and 10,000 cells per individual. Notably, we mixed cells from these five lines with cells from two additional lines (each with distinct genotypes) from a separate experiment during 10x collections. Libraries were sequenced using paired-end 100 bp sequencing on the NovaSeq 6000 at the University of Chicago Genomics Core. We aligned samples to the human genome (GRCh38) using *CellRanger* (*Zheng et al., 2017*). We then assigned cells to individuals. We used *demuxlet* to identify doublets with previously collected and imputed genotype data for the five additional YRI individuals; this data originated from the HapMap and 1000 Genomes Projects (*Kang et al., 2018*; *Auton et al., 2015*; *Belmont et al., 2003*). Finally, we removed cells assigned to individuals that were not a part of this experiment. We filtered out doublets, cells with greater than 15% mitochondrial reads or fewer than 3% mitochondrial reads, and cells with fewer than 1000 unique genes expressed.

We then normalized data using *SCTransform* (*Butler et al., 2018*; *Hafemeister and Satija, 2019*), identified clusters using the Louvain algorithm in Seurat (at Resolution 0.15), and visualized expression of canonical marker genes and the most significant marker genes of clusters identified in differential expression analysis (*Figure 1—figure supplement 4*). Based on marker gene expression, clusters 0 and 2 represent early ectoderm, cluster one represents neural crest cells, cluster three represents

pluripotent cells, clusters 4 and 6 represent neurons, cluster 5 represents mesoderm, cluster seven represents endoderm, and cluster 8 represents endothelial cells. We calculated the proportion of cells from each individual that were assigned to each of these cell type categories. We observed that each of these five additional cell lines exhibits high differentiation efficiency, comparable to that of iPSC lines 18511 and 19160. Additional lines were also integrated with reference data to annotate cell types as described below.

## Reference integration and label transfer

We next compared cells to reference data sets of primary fetal cell types, Day 20 hESC-derived EBs, and hESCs (*Cao et al., 2020*; *Han et al., 2020*). To integrate our cells with the reference sets, we first subset each reference set to include only protein coding genes. Because the Cao et al. reference set is so large, containing over 4 million cells, we subset cells from this reference set to include a maximum of 500 cells per cell type. We then normalized each reference set using SCTransform (*Butler et al., 2018*; *Hafemeister and Satija, 2019*). We then merged the data sets using Seurat, re-ran SCTransform regressing out data set specific effects of sequencing depth, scaled the data, and ran PCA. We then ran Harmony to correct PCA embeddings for the effects of each data set to complete the integration (*Korsunsky et al., 2019*). We then transferred cell type annotations from cell types present in the fetal reference and hESC to EB cells. For each EB cell, we found the five nearest reference cells in Harmony-corrected PCA space based on Euclidean distance; if at least 3/5 of the nearest reference cells shared an annotation, that annotation was transferred to the EB cell. If fewer than three of the nearest reference cells shared an annotation, the EB cell was annotated as 'uncertain'.

To assess the quality of our reference integration strategy, we asked whether (1) datasets are being over-corrected and (2) EB cells annotated using reference cell types express expected marker genes. We first subsetted EB cells to broad cell type categories identified using clustering (at resolution 0.1) and differential expression analysis: Pluripotent (cluster 0), Early Ectoderm (cluster 1), Endoderm (cluster 4), Meso-derm (clusters 2, 6), Neural Crest (cluster 3), and Neurons (cluster 5). Using each subset of cells, we repeated the reference integration pipeline by merging the EB cells with three reference data sets (fetal cells, hESCs, and an external set of Day 20 EBs), normalizing using SCTransform, running PCA using 5,000 variable features, in-tegrating the data using Harmony, and transferring labels based on the five nearest reference cells (see Materials and methods) (*Butler et al., 2018*; *Hafemeister and Satija, 2019*; *Korsunsky et al., 2019*). We found that 79% of EB cells are assigned to the same cell type in the full integration and subset integration. Of EB cells that are anno-tated differently in the full and subset integrations, 82% were labeled as 'hESC' or 'uncertain' in either the full or subset integration. This suggests that differences in these annotations are often be due to slight changes in the positioning of cells between the hESC reference cells and fetal reference cells; this is expected when pluripotent cells are not included in subsets of EB cells. And, importantly, cells are not often annotated as a different fetal cell type. Together, these results suggest that our integration approach is robust to subsetting input cell types and is likely not over-integrating the test and reference data sets.

Next, we asked whether annotated EB cells differentially express expected marker genes. We limited this analy-sis to annotations with at least 10 total EB cells from at least two individuals in two replicates. We then calculat-ed pseudobulk expression for cells of the same annotation, individual, and replicate, and filtered genes to in-clude only those with at least 10 counts in at least one sample and at least 15 total counts across all samples. We then TMM-normalized pseudobulk expression values, used voom to calculate a weighted gene expression value, and tested for differential expres-sion between annotations using limma. Of the annotations tested, the most significantly differentially expressed genes often included known cell type markers. For example, cells annotated as cardiomyo-cytes showed significant upregulation of of MYL7, MYL4, and TNNT2 (*Figure 2—figure supplement 3*). Cells annotated as hepatoblasts showed significant upregulation of AFP, FGB, and ACSS2. Cells annotated as mesothelial cells showed significant upregulation of NID2 and collagen genes (COL6A3, COL1A1, COL3A1, COL6A1). These results provide further support that our reference integration approach yields meaningful annotation of EB cells.

## Topic modeling

We also performed topic modeling using *FastTopics* to learn major patterns in gene expression within the data set, or topics, and model each cell as a combination of these topics. For this analysis, we

used raw counts and filtered the data to include genes expressed in at least 10 cells. We then pre-fit a Poisson non-negative matrix factorization by running 1000 EM updates without extrapolation to identify a good initialization at values of k equal to 6, 10, 15, 25, and 30. We used this initialization to fit a non-negative matrix factorization using 500 updates of the *scd* algorithm with extrapolation to identify 6, 10, 15, 25, and 30 topics. We then used *FastTopics'* *diff_count_analysis* function to identify genes differentially expressed between topics. We used these differentially expressed genes to interpret the cellular functions and identities captured by each topic. In some cases, differentially expressed genes included known marker genes (*Table 2*).

## Hierarchical clustering based on cell type composition and gene expression

To understand how similar cell type composition is between replicates and individuals, we first calculated the proportion of cells from each individual in each replicate assigned to each Seurat cluster at resolution 0.1. Then, using the *ComplexHeatmap* R package and performing hierarchical clustering based on the complete linkage method, we visualized the clustering of these replicate-individual groups (*Gu et al., 2016*). We repeated this analysis using Seurat clusters at resolution 0.5, 0.8, and 1 to show that the overall patterns of hierarchical clustering are robust to cluster resolution. We performed an analogous analysis using topic loadings instead of cluster proportions. Here, we determined the loading of each topic on cells from the same individual and replicate, then used the same hierarchical clustering with *ComplexHeatmap* to visualize patterns of similarity between cells of each individual and replicate (*Gu et al., 2016*).

We also performed hierarchical clustering on gene expression of individual cells. To do so, we took the pseudobulk expression for each individual-replicate-cluster group and filtered to genes expressed in at least 20% of cells in at least one cluster. We then calculated the log10 counts per million expression of each gene. We then generated a heatmap using the *ComplexHeatmap* R package, again performing hierarchical clustering based on the complete linkage method (*Gu et al., 2016*).

## Variance partitioning

Using the same pseudobulk data and precision weights computed by *voom* from differential expression analysis, we used the *VariancePartition* R package to quantify the variation attributable to cluster, replicate, and individual (*Hoffman and Schadt, 2016*). We fit a random effect model and modeled cluster, replicate, and individual as random effects. We performed this analysis across all tested Seurat clustering resolutions (0.1, 0.5, 0.8, 1). We performed this analysis using both pseudobulk samples of cells from the same cluster, replicate, and individual and at single-cell resolution with each cell as a sample. We also partitioned the variance in each Seurat cluster separately using a random effect model with terms for replicate and individual. For this analysis, we used pseudobulk samples of cells from each individual and replicate.

## Power analysis

To ascertain the power to detect eQTLs and dynamic eQTLs across a range of sample sizes, standardized effect sizes, and experiment sizes we used a power function as derived in *Sarkar et al., 2019*:

$$Pow\left(\beta, \alpha, n, \sigma\right) = \Phi\left(\Phi^{-1}\frac{\alpha}{2} + \frac{\beta}{\sigma}\sqrt{n}\right)$$

where β denotes the true additive significance level, α denotes the significance level, n denotes sample size, and σ represents the phenotype standard deviation. This approach assumes a simple linear regression for eQTL mapping and a conservative Bonferroni correction (FWER = 0.05, assuming 10,000 genes tested, $\alpha$ = 5e-6).

To estimate the standard deviation for a given experiment size, we downsampled cells from this experiment, sampling evenly between individuals and replicates to range of experiment sizes from 2700 cells to 21,600 cells. For each experiment size, we took 10 random samples of cells and calculated pseudobulk expression of cells from the same cluster (defined at resolution 1), individual, and replicate. We filtered to include genes expressed in at least 20% of cells in at least one cluster (at resolution 1) in the full set of EB cells. For each sample we then partitioned the median variance attributable to residuals using the *variancePartition* package. We then took the median of the median

variance from the 10 samples at each experiment size and fit an exponential decay model to quantify the relationship between experiment size and residual variance. We used square root of this variance to determine the standard deviation for a given experiment size in our power calculations.

## Trajectory inference and identification of dynamic gene modules

We inferred trajectories using PAGA in Scanpy using Seurat clusters at all tested resolutions (*Wolf et al., 2019*). We assigned pseudo-time using diffusion pseudo-time with the pluripotent cells assigned as the root (*Haghverdi et al., 2016*). We then manually traced known developmental trajectories supported by the PAGA graph. At clustering resolution 1, we traced the trajectory from pluripotent cells to hepatocytes (clusters 0, 1, 5, 6, 7, 10, 11, 16, 18, 19, 25, and 22), pluripotent cells to endothelial cells (clusters 0, 1, 4, 5, 6, 7, 11, 16, 18, 22, 24, and 25), and pluripotent cells to neurons (clusters 0, 1, 2, 3, 5, 6, 7, 8, 9, 11, 12, 13, 14, 15, 16, 17, 18, 20, 21, 26, and 27) (*Figure 7A–C*).

We then isolated cells from each of these three trajectories and used Split-GPM to simultaneously cluster samples and identify dynamic gene modules. For this analysis, we divided data into decile pseudo-time bins and calculated pseudobulk gene expression for cells of the same individual, replicate, and pseudo-time bin. We identified 20 dynamic gene modules in each trajectory and interpreted them using gene set enrichment. To understand the variation in dynamic gene expression between individuals and replicates, we re-ran split-GPM ten times and observed how often cells from each individual and replicate were assigned to the same sample cluster.

## Acknowledgements

We thank all members of the Gilad lab, Sebastian Pott, and Abhishek Sarkar for helpful discussions, and Natalia Gonzales for comments on the manuscript. We thank Jay Shendure, Silvia Domke, and Sam Regalado for their helpful discussion of the reference integration and cell type annotation analyses. We thank Sandhiya Arun and the Genomics Core Facility at the University of Chicago for performing 10x collections and sequencing the libraries. YG was funded by NIH grant R35GM131726. KR was supported by F31HL146171. AB was funded by NIH grant R35GM139580. The computational resources were provided by the University of Chicago Research Computing Center.

## Additional information

### Competing interests

Katherine Rhodes: KR is named as an inventor with the University of Chicago on a patent related to the this manuscript (patent pending 63/291,945). Kenneth A Barr: KAB is named as an inventor with the University of Chicago on a patent related to the this manuscript (patent pending 63/291,945). Alexis Battle: AB is a consultant for Third Rock Ventures, LLC and a shareholder in Alphabet, Inc. Yoav Gilad: YG is named as an inventor with the University of Chicago on a patent related to the this manuscript (patent pending 63/291,945). The other authors declare that no competing interests exist.

### Funding

| Funder | Grant reference number | Author |
| --- | --- | --- |
| National Heart, Lung, and Blood Institute | Ruth L. Kirschstein National Research Service Award Individual Predoctoral Fellowship F31HL146171 | Katherine Rhodes |
| National Institute of General Medical Sciences | R35GM131726 | Yoav Gilad |
| National Institute of General Medical Sciences | R35GM139580 | Alexis Battle |

The funders had no role in study design, data collection and interpretation, or the decision to submit the work for publication.

## Author contributions
Katherine Rhodes, Conceptualization, Data curation, Formal analysis, Investigation, Methodology, Visualization, Writing – original draft, Writing – review and editing; Kenneth A Barr, Data curation, Formal analysis, Investigation, Methodology, Writing – review and editing; Joshua M Popp, Data curation, Formal analysis, Visualization, Writing – review and editing; Benjamin J Strober, Formal analysis, Writing – review and editing; Alexis Battle, Yoav Gilad, Conceptualization, Funding acquisition, Resources, Supervision, Writing – review and editing

## Author ORCIDs
Katherine Rhodes ⓘ http://orcid.org/0000-0002-0631-3994
Kenneth A Barr ⓘ http://orcid.org/0000-0002-0769-7053
Alexis Battle ⓘ http://orcid.org/0000-0002-5287-627X
Yoav Gilad ⓘ http://orcid.org/0000-0001-8284-8926

## Decision letter and Author response
Decision letter https://doi.org/10.7554/eLife.71361.sa1
Author response https://doi.org/10.7554/eLife.71361.sa2

## Additional files

### Supplementary files
• Supplementary file 1. Frequency of each cell type present in EB data from lines 18511, 18858, and 19,160 after transferring annotations from the fetal and hESC reference sets.

• Supplementary file 2. Frequency of each cell type present in EB data from additional lines after transferring annotations from the fetal and hESC reference sets.

• Supplementary file 3. Number of cells per cluster (resolution 0.1) from each individual-replicate sample.

• Supplementary file 4. Limma differential expression results (logFC, AveExpr, t, P.Value, adj.P.Val, B) for all tested genes in each Seurat cluster when clusters were defined at resolution 0.1.

• Supplementary file 5. Limma differential expression results (logFC, AveExpr, t, P.Value, adj.P.Val, B) for all tested genes in each Seurat cluster when clusters were defined at resolution 0.5.

• Supplementary file 6. Limma differential expression results (logFC, AveExpr, t, P.Value, adj.P.Val, B) for all tested genes in each Seurat cluster when clusters were defined at resolution 0.8.

• Supplementary file 7. Limma differential expression results (logFC, AveExpr, t, P.Value, adj.P.Val, B) for all tested genes in each Seurat cluster when clusters were defined at resolution 1.

• Transparent reporting form

### Data availability
Sequencing Data have been deposited in GEO under accession code GSE178274. Code used in this project is available on GitHub: Workflowr site: Workflowr site: https://klrhodes.github.io/Embryoid_Body_Pilot_Workflowr/index.html, Preprocessing: https://github.com/kennethabarr/HumanChimp, (copy archived at swh:1:rev:1722b3484d1120384c910bef13bffd6a6cd6179a), Additional Preprocessing, Integration, Differential Expression, Topic Modelling, Variance Partitioning, Hierarchical Clustering, and Reference Annotation: https://github.com/KLRhodes/Embryoid_Body_Pilot_Workflowr, (copy archived at swh:1:rev:3d3fa9b4e0a96ea840009b62107d47e464e08c11) Trajectory Inference, Identification of dynamic gene modules: https://github.com/jmp448/ebpilot, (copy archived at swh:1:rev:8b9b9700063610b4f9f2406131b646542bfa7af2).

The following dataset was generated:

| Author(s) | Year | Dataset title | Dataset URL | Database and Identifier |
| --- | --- | --- | --- | --- |
| Rhodes K, Barr KA, Popp JM, Strober BJ, Battle A, Gilad Y | 2021 | Human embryoid bodies as a novel system for genomic studies of functionally diverse cell types | https://www.ncbi.nlm.nih.gov/geo/query/acc.cgi?acc=GSE178274 | NCBI Gene Expression Omnibus, GSE178274 |

The following previously published datasets were used:

| Author(s) | Year | Dataset title | Dataset URL | Database and Identifier |
|-----------|------|---------------|-------------|-------------------------|
| Han X, Guo G, Zhou Z, Fei L, Sun H, Wang R, Wang J, Chen H | 2020 | Construction of a human cell landscape at single-cell level | https://www.ncbi.nlm.nih.gov/geo/query/acc.cgi?acc=GSE134355 | NCBI Gene Expression Omnibus, GSE134355 |
| Cao J, O'Day DR, Pliner HA, Kingsley P, Deng M, Daza RM, Zager MA, Kimberly A, Blecher R, Zhang F, Spielmann M, Palis J, Doherty D, Steemers FJ, Glass IA, Trapnell C, Shendure J | 2020 | A human cell atlas of fetal gene expression | https://www.ncbi.nlm.nih.gov/geo/query/acc.cgi?acc=GSE156793 | NCBI Gene Expression Omnibus, GSE156793 |

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
