## [Editor Report]

The authors generated embryoid bodies (EBs) from induced pluripotent stem cells (iPSCs) using a strong mixed-pool study design and performed scRNA-seq profiling. From this data, they identify dozens of cell types and infer differentiation trajectories that align well with known developmental gene expression dynamics. This system is likely to be a good platform for larger eQTL studies that interrogate new cell states.

---

## [Decision Letter]

**Decision letter after peer review:**

Thank you for submitting your article "Human embryoid bodies as a novel system for genomic studies of functionally diverse cell types" for consideration by *eLife*. Your article has been reviewed by 2 peer reviewers, one of whom is a member of our Board of Reviewing Editors, and the evaluation has been overseen by a Reviewing Editor and Patricia Wittkopp as the Senior Editor. The reviewers have opted to remain anonymous.

Essential revisions:

1) There seems to be a disconnect between what the study should be about and what it is about. Consequently, the message is a bit confusing, and the relevance of the findings is not so clear. The abstract and introduction are both focused on the potential use of EBs for the mapping of genetic variants affecting gene expression. However, the current manuscript is not an eQTL mapping study, but rather a characterization of EBs by single cell RNA-seq. It would be important to write the abstract, Introduction and title in order to make them more relevant to the data presented. For example, a major difficulty for the authors is the identification of different cell types, this should be discussed in the Introduction, rather than the difficulty in mapping genetic variants.

2) line 104: "we found that every replicate in our experiment, regardless of the individual, includes cells from all three germ layers (Figure 1E, Figure 1G)".

Actually, Figure 1G shows that all 18858 samples contain almost exclusively pluripotent cells (cluster 0). More in general, the characterization of the embryoid bodies used for the study is insufficient. The entire study is based on the analyses of embryoid bodies collected after 3 weeks, generated from 3 iPS cell lines. However, 1 line has clear differentiation defects, or that embryoid bodies differentiation experiment was problematic.

Although this reviewer appreciates that the focus of the study is the generation and analysis of single cell data, if the biological material analysed is of poor quality, the entire study could be questioned. To address this point the authors should perform additional experiments (immunostaining, qPCR) at different days of embryoid body differentiation to make sure that the procedure is correct, that all lines can efficiently differentiate. If they confirm with independent techniques that the 18858 line is almost unable to differentiate, I would suggest to remove it from the analyses, as it simply is a faulty line that cannot be called pluripotent and would not be used by others. For example, the conclusions of lines 324-325 "Replicates of 18858 often cluster together and rarely clustered with the other individuals, suggesting that not only did 18858 have poor differentiation efficiency, but cells that did differentiate show a distinct pattern of expression dynamics" would suggest that such line is problematic and should not be used.

3) With so few samples multiplexed together, demuxlet may not be able to detect all likely doublets. It may be helpful to run an independent doublet finder on the pass-QC cells to make sure there are no same-sample doublets (that demuxlet would miss).

*Reviewer #1:*

Rhodes and colleagues generated a dataset of single cell RNA-seq of embryoid bodies from 3 iPS cell lines and analysed them with the aim of identifying different cell type and also to partition the variance among biological and technical. The authors propose also to use in the future such system to map genetic variants affecting gene expression, but this interesting aspect is not fully explored in the current manuscript.

The different strategies used to identify different cell types in the dataset are very interesting and could be used for the study of organoids or other tissues by single cell RNA-seq.

The data and analyses pipelines could be useful to other colleagues, but some points need to be clarified.

In line 104: "we found that every replicate in our experiment, regardless of the individual, includes cells from all three germ layers (Figure 1E, Figure 1G)".

Actually, Figure 1G shows that all 18858 samples contain almost exclusively pluripotent cells (cluster 0). More in general, the characterization of the embryoid bodies used for the study is insufficient. The entire study is based on the analyses of embryoid bodies collected after 3 weeks, generated from 3 iPS cell lines. However, 1 line has clear differentiation defects, or that embryoid bodies differentiation experiment was problematic.

Although this reviewer appreciates that the focus of the study is the generation and analysis of single cell data, if the biological material analysed is of poor quality, the entire study could be questioned.

*Reviewer #2:*

Here the authors generated embryoid bodies (EBs) from iPSCs from three individuals, with three replicates each, and performed scRNA profiling yielding a total of 42,488 cells. The QC performed to get this collection of cells is strong, especially in using the demuxlet approach. From this data, they identify dozens of cell types and infer differentiation trajectories that align well with known developmental gene expression dynamics. They propose this system is therefore an ideal platform for larger eQTL studies.

Much of the introduction motivates this study by using GWAS and eQTL studies and their corresponding gaps. The idea is to capture unexplored cell states using the EB model and in a later study scale up to enable eQTL scans. Thus, the EB platform should represent convincingly novel cell state space compared to existing data. The pseudotime results seem to indicate this with concordance across reference markers for three different lineages. The power calculations are nice to see, but are an exercise in setting expectations for a future study. Overall, this is a strong paper that represents the initial survey of a potentially exciting eQTL platform.

[Editors' note: further revisions were suggested prior to acceptance, as described below.]

Thank you for resubmitting your work entitled "Human embryoid bodies as a novel system for genomic studies of functionally diverse cell types" for further consideration by *eLife*. Your revised article has been evaluated by Patricia Wittkopp (Senior Editor) and a Reviewing Editor.

The manuscript has been improved but there are some remaining issues that need to be addressed, as outlined below:

The addition of five more samples to bolster the conclusions observed in the initial three samples is good. However, these additional five samples are not mentioned in the introduction (only the original three are mentioned) and the new results are only presented in supplementary figures. This presents the odd scenario in which the main figures depict less samples (three) than the supplementary figures (five). We therefore recommend combining data from all samples and presenting in main figures, whenever possible. Further, the GEO accession (GSE178274) looks like it only includes the initial three samples. We recommend sharing all data and request that the GEO accession be updated to include the five additional samples (all eight samples total).

---

## [Author Response]

Essential revisions:1) There seems to be a disconnect between what the study should be about and what it is about. Consequently, the message is a bit confusing, and the relevance of the findings is not so clear. The abstract and introduction are both focused on the potential use of EBs for the mapping of genetic variants affecting gene expression. However, the current manuscript is not an eQTL mapping study, but rather a characterization of EBs by single cell RNA-seq. It would be important to write the abstract, Introduction and title in order to make them more relevant to the data presented. For example, a major difficulty for the authors is the identification of different cell types, this should be discussed in the Introduction, rather than the difficulty in mapping genetic variants.

We accept this comment. The abstract and introduction have been rewritten to focus on the results of the current experiment, including challenges in classifying cell types. We believe that the title is appropriate for both the results presented and the conclusions drawn from this study.

2) line 104: "we found that every replicate in our experiment, regardless of the individual, includes cells from all three germ layers (Figure 1E, Figure 1G)".Actually, Figure 1G shows that all 18858 samples contain almost exclusively pluripotent cells (cluster 0). More in general, the characterization of the embryoid bodies used for the study is insufficient. The entire study is based on the analyses of embryoid bodies collected after 3 weeks, generated from 3 iPS cell lines. However, 1 line has clear differentiation defects, or that embryoid bodies differentiation experiment was problematic.Although this reviewer appreciates that the focus of the study is the generation and analysis of single cell data, if the biological material analysed is of poor quality, the entire study could be questioned. To address this point the authors should perform additional experiments (immunostaining, qPCR) at different days of embryoid body differentiation to make sure that the procedure is correct, that all lines can efficiently differentiate. If they confirm with independent techniques that the 18858 line is almost unable to differentiate, I would suggest to remove it from the analyses, as it simply is a faulty line that cannot be called pluripotent and would not be used by others. For example, the conclusions of lines 324-325 "Replicates of 18858 often cluster together and rarely clustered with the other individuals, suggesting that not only did 18858 have poor differentiation efficiency, but cells that did differentiate show a distinct pattern of expression dynamics" would suggest that such line is problematic and should not be used.

Our brief response: To address this, we collected and now present EB single cell data from 5 more individuals to confirm that 18858 is an exception. See below.

More detailed response:

The pluripotency of all lines used in this study, including 18858, has been previously validated using several functional assays (Banovich et al., 2018). At the onset of our response to this concern we wish to state that we feel that it would be inappropriate to exclude data from 18858 from this study. The less efficient differentiation of this line provides a real perspective on the utility of our new model, and we do not wish to exclude this line and risk providing a too-optimistic view of the model.

With that in mind, the statement that “all 18858 samples contain almost exclusively pluripotent cells (cluster 0)” is not correct. Our single cell data set is consistent with the conclusion that 18858 is pluripotent and produces cells of all three germ layers, including mature cell types. It is correct that we observe much lower differentiation efficiency with this line. To visualize the cell type composition of 18858 EBs more clearly, we have added supplement figure S5. Additionally, the number of cells assigned to each cluster (resolution 0.1) from each replicate of each individual is shown in original table S2. While 89% of EB cells from 18858 have remained pluripotent, this individual does produce cells assigned to endoderm (cluster 4, Resolution 0.1), mesoderm (clusters 2 and 6, Resolution 0.1), and ectoderm (clusters 1 and 5, Resolution 0.1). While only a relatively small proportion of EB cells are assigned to these clusters compared to the pluripotent cluster (cluster 0, Resolution 0.1), the cells present in each are high quality and the quantities of cells are sufficient for our analyses of biological and technical variation. More detail about the cell type composition of this line has also been added to the main text (lines 174-183).

Perhaps the larger point made by the reviewer is that with only 3 individuals, one that did not differentiate as efficiently, we are unable to determine whether poor differentiating EBs will be common or rare. This point, we entirely accept and to address it, we had to collect additional data.

Specifically, we generated EBs from 5 additional YRI iPSC lines (18856, 18912, 19140, 19159, and 19210). A single replicate of these lines were differentiated in parallel, dissociated in parallel, and then mixed in equal proportions prior to single cell collection using the 10x (see updated Methods section, lines 753-777). Supplemental figure S6 shows quality metrics associated with the data collected from these new EBs. After filtering and normalizing, we clustered the cells and annotated clusters using both canonical markers of each germ layer as well as marker genes learned from differential expression analysis in the original 3 lines. These new lines show the presence of similar cell types to those observed in the original dataset; coarse clustering of these cells reveals clusters representing pluripotent cells, endoderm, mesoderm, early ectoderm, neural crest, neurons, and endothelial cells. All 5 of the newly collected lines have proportions of pluripotent cells under 25%, indicating these lines had differentiation efficiencies that more closely resembled those of 18511 and 19160 in the original data set. Overall, the cell type composition of these new lines suggests that most iPSC lines from the YRI panel will differentiate well using this protocol.

3) With so few samples multiplexed together, demuxlet may not be able to detect all likely doublets. It may be helpful to run an independent doublet finder on the pass-QC cells to make sure there are no same-sample doublets (that demuxlet would miss).

Brief response: We accepted the comment and tried an independent doublet finder; it did not do better and we believe we understand the reason for this.

More detailed response:

We agree that if we had included only 3 individuals in the 10x collections, the number of samples would be too small to use demuxlet alone for doublet detection. However, as described in the “Single cell sequencing” Methods section (lines 455-463), these 10x collections included samples from additional lines that were not analyzed in this study, but were used for doublet detection with demuxlet. In total, the first batch of collections (including all “replicate 1” samples), included a total of 9 lines from different individuals. Given our loading of 10,000 cells into each 10x lane, we expect a multiplet rate of 7.6%. With 9 individuals, we expect 11.1% of all doublets to be from the same individual. This would suggest that 0.84% of batch 1 cells are doublets missed by demuxlet (141 cells total that passed QC). Batches 2 and 3 of collections (containing “replicate 2” and “replicate 3” samples, respectively), each included a total of 6 lines from different individuals. With 6 individuals, we expect that 16.6% of doublets are from the same individual; this suggests that 1.27% (or 325 cells that passed QC) from batches 2 and 3 are doublets missed by demuxlet. In total, we expect only 466 same-individual cells. Of these, some will represent ‘homotypic’ doublets—cells in a similar transcriptional state. Overall, the proportion of same-individual heterotypic doublets is relatively small.

We agree, however, that these same-individual heterotypic doublets should be removed if they can be reliably identified by other methods. Most doublet-calling methods able to identify same sample doublets use a similar procedure involving the generation of artificial doublets from cells in the data and comparison of cells to those artificial doublets. We were concerned that in the EB data set, which includes many intermediate, developing cell types, these intermediate cell types may tend to be erroneously called as doublets. We tested the performance of one such method, DoubletFinder (McGinnis et al., 2019), on cells from a single 10x lane. Using Seurat, we first filtered cells using the same criteria as the original data (see Filter and integration Methods), but kept known doublets called by demuxlet, but removed both singlets and doublets not assigned to individuals used in this study. We then normalized data using SCTransform, ran PCA, clustered cells (Resolution 0.8), and ran DoubletFinder. This lane of 10x cells included 2.5% known doublets called by demuxlet (after filtering) and contains a mix of 6 individuals (from a batch 2 collections); we therefore expect 1.27% of cells to be same-sample doublets. We therefore assigned expected doublet rate of 3.8% in DoubletFinder.

We found that of 64 known doublets called by demuxlet, only 13 were identified as doublets by DoubletFinder. Because DoubletFinder misses over 75% of known doublets in this dataset, it is likely missing true same-sample doublets as well. Moreover, cells called as doublets by doublet finder are concentrated in the area of the UMAP representing early differentiating cells, while the known doublets are distributed more evenly across all cell clusters, as shown in Author response image 1. This discrepancy between the transcriptional state of known doublets compared to inferred doublets suggests that DoubletFinder, and other similar methods, cannot reliably call doublets in datasets with continuous cell state gradients. We therefore chose not to remove the DoubletFinder-inferred doublets from our dataset and proceed with the knowledge that a small percentage (less than 1.1%) of cells in our data likely represent same-individual heterotypic doublets.

**Author response image 1. sa2fig1:** Same-Sample doublet detection test. (**A**) Table comparing the demuxlet cell assignments to the DoubletFinder assignments. (**B**) UMAP plot showing cells colored by normalized expression of marker genes for pluripotency (POU5F1), early ectoderm (PAX6), mesoderm (HAND1), and endoderm (SOX17). (**C**) UMAP plot showing cells colored by DoubletFinder’s pANN (proportion of artificial k nearest neighbors) metric, where cells with the highest pANN are assigned as doublets based on a given threshold. (**D**) UMAP plot showing cells colored by DoubletFinder assignment (**D**) UMAP plots split by cells of each demuxlet assignment and colored by cluster assignment as resolution 0.8.

References

Banovich, N. E., Li, Y. I., Raj, A., Ward, M. C., Greenside, P., Calderon, D., Tung, P. Y., Burnett, J. E., Myrthil, M., Thomas, S. M., Burrows, C. K., Romero, I. G., Pavlovic, B. J., Kundaje, A., Pritchard, J. K., & Gilad, Y. (2018). Impact of regulatory variation across human iPSCs and differentiated cells. *Genome Res.*, *28*(1), 122–131. https://doi.org/10.1101/gr.224436.117

McGinnis, C. S., Murrow, L. M., & Gartner, Z. J. (2019). DoubletFinder: Doublet Detection in Single-Cell RNA Sequencing Data Using Artificial Nearest Neighbors. *Cell Systems*, *8*(4), 329-337.e4. https://doi.org/10.1016/J.CELS.2019.03.003

[Editors' note: further revisions were suggested prior to acceptance, as described below.]

The manuscript has been improved but there are some remaining issues that need to be addressed, as outlined below:The addition of five more samples to bolster the conclusions observed in the initial three samples is good. However, these additional five samples are not mentioned in the introduction (only the original three are mentioned) and the new results are only presented in supplementary figures. This presents the odd scenario in which the main figures depict less samples (three) than the supplementary figures (five). We therefore recommend combining data from all samples and presenting in main figures, whenever possible.

It was not trivial to address this, as the study goals and the way we structured the results were focused on the nested property of our original study design. We could not actually perform a combined analysis of data from all 8 individuals in all sections of the paper. Because we collected a single replicate of the new lines to address a very specific reviewer’s concern in the previous round, the utility of these new data, in the context of our study, is limited to the exploration of cell type composition.

With that in mind, we added details about the collection of all 8 lines to the study design section at the beginning of the results, as requested, and moved the stacked barplot showing cell type composition of the additional lines to the main text (now figure 1H). We have also expanded our analyses of the new lines by integrating these cells with the fetal and hESC reference sets to determine the consistency reference cell type annotation (Figure 3 and corresponding supplementary figures).

Further, the GEO accession (GSE178274) looks like it only includes the initial three samples. We recommend sharing all data and request that the GEO accession be updated to include the five additional samples (all eight samples total).

Thank you. The GEO accession has been updated to include all samples.